# Iterative Improvements Based on Ground Truth: Building LLM Agents in the Era of Experience Inspired by Games AI

## Abstract

LLM agents have attracted much attention recently. However, how to build successful LLM agents, esp. w.r.t. autonomy and optimality, is still an open problem. We present a perspective paper with a brief survey about building LLM agents with iterative improvements based on ground truth, in the era of experience inspired by the successes of games AI. We propose AgentZero, Agent$\mu$, and Agent$\infty$, agent frameworks with perfect, learned and no world models, following AlphaZero, MuZero and a model-free method like DQN, respectively. We propose to leverage domain knowledge for data collection, architecture design and algorithm design, and propose decision time planning and meta reinforcement learning at both pre- and post-training stages. We present case studies for building agents for games, maths, or coding, with approximate simulators, facts, and/or human-in-the-loop.

## 1 Introduction

The whole fields of artificial intelligence (AI) (Russell & Norvig, 2020) and reinforcement learning (RL) (Sutton & Barto, 2018) are constructed around agents: relevant techniques to make agents feasible/optimal, from prediction or machine learning methods for function approximations with e.g. neural networks or decision trees, to decision-making frameworks like search, planning and reinforcement learning, and to fields for perception (computer vision), communication (natural language processing) and acting (robotics). Silver & Sutton (2025) draw a big picture of the era of simulation, era of human data and era of experience. Alan Turing remarked in 1947 "What we want is a machine that can learn from experience."

Large language models (LLMs) have been making significant progress, e.g., BERT (Devlin et al., 2019), T5 (Raffel et al., 2020), GPT-3 (Brown et al., 2020), ChatGPT (OpenAI, 2022), GPT-4 (OpenAI et al., 2024a), Claude (Anthropic, 2024), Gemini (Gemini Team et al., 2024), Llama (Grattafiori et al., 2024), OpenAI o1 (OpenAI et al., 2024b), DeepSeek-V3 (DeepSeek-AI, 2024), and DeepSeek-R1 (DeepSeek-AI, 2025). LLM-based agents attract lots of attention recently (Kapoor et al., 2024).

Games AI boasts many successful agents, e.g., AlphaGo (Silver et al., 2016), based on which, we can build (close to) autonomous and optimal game programs. Games AI has changed the games industry and community, e.g., chess players have to learn from chess programs. However, how to build successful LLM agents, esp. w.r.t. autonomy and optimality, is still an open problem (Morris et al., 2024; Feng et al., 2024). LLM agents are still relying on manual and/or heuristic (approximate) methods, e.g., chain of thought (CoT) (Wei et al., 2022; Stechly et al., 2024b), ReAct (Yao et al., 2023; Verma et al., 2024), workflows, and imperfect world models (Vafa et al., 2024; Wang et al., 2024c; Li et al., 2025). A root cause is the fixed yet imperfect LLMs. Games AI features iterative improvements based on ground truth. However, most LLM agents lack frequent iterative improvements, esp. for the underlying LLMs, and many LLM agents lack ground truth. Moreover, games are broad, with a wide range of problem formulations and applications, considering also game theory (Leyton-Brown & Shoham, 2008) and gamification (Werbach & Hunter, 2020), characterizing many dimensions, like discrete or continuous for time, observations and actions, deterministic or stochastic for dynamics, and single or multi-agent, similar to RL (Silver et al., 2021). Games to AI is like fruit flies to genetics. We expect games AI inspires further progress in LLM agents.

We present a perspective paper with a brief survey about building LLM agents with iterative improvements based on ground truth, in the era of experience, inspired by the successes of games AI. We propose AgentZero and Agent$\mu$, agent frameworks with perfect and learned world models, following AlphaGo and MuZero, respectively. We also propose a model-free framework Agent$\infty$. We propose to leverage domain knowledge for data collection, architecture design and algorithm design, and propose decision time planning and meta reinforcement learning at both pre- and post-training stages. We present case studies for building agents for games, maths, or coding, with approximate simulators, facts, and/or human-in-the-loop.

**What are ground truth?** We identify three sources of ground truth data: 1) perfect verifiers with formal methods, 2) perfect rules/laws/world models, and 3) experience by interacting with the world.[1]

Sources 1 and 2 can guarantee correctness, e.g., AlphaZero (Silver et al., 2018) with a perfect game rule, and AlphaGeometry (Trinh et al., 2024) with a perfect theorem prover. Facts are in Source 2. We treat this as ground truth in the strong sense. Note, knowledge may evolve; see Section 4. For Source 3, the world can be viewed as a perfect model itself. However, Source 3 may come with noises or uncertainty. We may need to deal with sampling errors, partial observability, imperfect/incomplete information, strategic/adversarial scenarios, and/or weak specifications like code unit tests. Also, collecting experience requires interactions with the world, which may be costly. Sources 1 and 2 are special cases of Source 3, with experience from interactions with special environments, like a theorem prover or a game engine. In practice, we may define ground truth in a broad sense, including code tests, annotated examples, environment feedback and experts' feedback, etc. That is, we treat Source 3 as ground truth in the broad sense.

We take the broad view in this paper, being aware that it may not guarantee correctness, e.g., as in formal verification. One factor is, ground truth in the strong sense may be limited, available only to a small classes of applications, e.g., games, maths and coding, while ground truth in the broad sense are necessary and valuable complement. We attempt to reconcile formal verification and LLMs communities. The strong sense, esp. Source 1 perfect verifiers with formal methods, corresponds to the formal verification community; see e.g. Li et al. (2024) and Yang et al. (2024a). The broad sense, which may correspond to weak specifications (Stoica et al., 2024) w.r.t. formal verification, is close to works in the LLMs community, which may take an LLM as a verifier. Note, mathematical proofs generated by LLMs are not treated as ground truth, before being verified with correctness guarantee. Also note, there are papers talking about "verifier" and "verifiable", but not in the sense of formal verification, or not ground truth in the strong sense, e.g., training a verifier (Cobbe et al., 2021), RL with verifiable rewards (RLVR) (Lambert et al., 2024), and generalization of RLVR (Su et al., 2025).

Silver & Sutton (2025) discuss the importance of experience from interacting with the real world and reinforcement learning for further progress in AI, at an abstract level. We provide a more pragmatic plan as a complement with a pseudo-code and discussions for a potential implementation. We highlight the importance of ground truth, in particular, in the strong sense with perfect models and formal verification, and the difficulties of generalizing / transferring achievements in games, maths, and coding to other problems. We also highlight that iterative improvements are a mechanism for agents to achieve grounding and agency. See also discussions about agency, e.g., Bisk et al. (2020) and Laird et al. (2023).

We discuss the following research questions (RQ). RQ1: Should we leverage domain knowledge? RQ1.1: How to leverage domain knowledge? RQ2: How to attain ground truth data? RQ3: How to make iterative improvements? RQ3.1: Why are iterative improvements important? RQ3.2: What are limitations of current LLMs? RQ3.3: Will scaling up current LLMs achieve reasoning and planning capacity to build agents? RQ3.4: Large vs small models? RQ3.5: Modularity? Generalist vs specialist? RQ3.6: What algorithms are suitable for iterative improvements?

This is a perspective paper. It is not a comprehensive survey. Considering the wide range of background knowledge in RL, LLMs, deep learning, and AI, as well as many topics we touch, like games, maths, coding, simulation, robotics, human-in-the-loop, etc., we do not attempt to make the paper fully self-contained. Otherwise, it would be very long. Instead, we expect a reader to have a decent background in relevant areas.

---

[1]We follow the definition of a (world) model as transition (dynamics) and reward models / functions in Sutton & Barto (2018). This is consistent with the definition of a world model in LeCun (2022). A model may also refer to a machine learning model or a language model. This should not be ambiguous in the context.

Our paper is about a blueprint for a potential, promising, near future solution. Our proposal is in stark contrast to the current popular approaches like LLM-based agents with prompt engineering and workflows. Ideally, a full implementation of our proposal should work at the pre-training stage, considering not only data collection and algorithm design, but also architecture design. That is, we may consider alternatives to (vanilla) Transformers and next token prediction. Different (neural network) architectures may be more suitable for different problems, as we discuss for RQ3.2 in Section 6. As a result, there are chances to build better base models, and make fine-tuning and prompting more efficient and effective. Thus our proposal is different from the current popular approaches about post-training. On the other hand, as a more "practical", interim solution, our proposal may be adapted to work in a post-training style. We hope to express our perspective so that there may be more resources allocated to approaches alternative to currently popular ones, in particular, LLM-based agents following the pipeline: 1) pre-training an LLM using GPT with next token prediction by some resource-rich organization, then, 2) relying on the fixed LLM, using prompts, workflows, etc. to build agents by the community.

We make the following contributions for building LLM agents: 1) We highlight the importance of both iterative improvements and ground truth. 2) We highlight the importance of domain knowledge, w.r.t. data collection, architecture design and algorithm design, and to strike a balance between a) scale and b) performance, efficiency and cost. 3) We expect to help improve the interpretability, trustworthiness and safety. 4) We briefly survey relevant works, make a critical examination, and propose a potential solution.

Next we introduce background in agents and games AI. We propose agent frameworks in Section 3. Then we discuss research questions about domain knowledge, ground truth and iterative improvements, in Section 4, 5 and 6, respectively. We discuss case studies in Section 7, compare with current approaches in Section 8, and close with a discussion.

## 2 Background

### 2.1 Agents

What is an agent? In Sutton & Barto (2018): "The learner and decision maker is called the agent." In Russell & Norvig (2020): "An agent is anything that can be viewed as perceiving its environment through sensors and acting upon that environment through actuators. For each possible percept sequence, a rational agent should select an action that is expected to maximize its performance measure, given the evidence provided by the percept sequence and whatever built-in knowledge the agent has." Decision making involves action selection, and is distinct from prediction, e.g., classification and regression by supervised learning. Note, however, imitation learning (Osa et al., 2018) applies supervised learning to decision making, in behavioral cloning and in inverse reinforcement learning (in the reward learning part). LLM agents leverage LLMs for building agents. We focus on multi-step, sequential decision making agents, with the goal of maximizing some performance metric. The aim is to build autonomous and optimal agents.

### 2.2 Games AI

AlphaGo (Silver et al., 2016), AlphaGo Zero (Silver et al., 2017), and AlphaZero (Silver et al., 2018) are characterized by perfect information w.r.t. the rule and the observability. MuZero (Schrittwieser et al., 2020) learns an iterable model that produces predictions relevant to planning: the action-selection policy, the value function and the reward. AlphaGo and MuZero correspond to model-based RL with a perfect and a learned model, respectively. Deep Q-Network (DQN) (Mnih et al., 2015) for Atari games is model-free RL, i.e., without a model. Note, however, there are perfect game engines in DQN. Dreamer (Hafner et al., 2025; 2020) for many games, including Minecraft, is a model-based method. Here a model means a world model.

AlphaGo series and MuZero use MCTS for decision time planning for lookahead search (Sutton & Barto, 2018). Experiments show the raw neural network, without MCTS after training, achieved an Elo rating of 3055 and AlphaGo Zero with MCTS at decision time achieved a rating of 5185 (Silver et al., 2017). Such results support the recent test time compute in LLMs. See e.g., OpenAI o1, Chen et al. (2024) and Wan et al. (2024). The results also show that imitation learning is not enough to achieve superhuman performance.

Besides a world model, we may refine agent design in other factors, e.g., multi-agent, imperfect information, and/or verbal communication. There are corresponding successful games AI, e.g., AlphaStar (Vinyals et al., 2019) for StarCraft, DeepStack (Moravčík et al., 2017), Libratus (Brown & Sandholm, 2017) and Pluribus (Brown & Sandholm, 2019) for Poker, and Cicero (Bakhtin et al., 2022) for Diplomacy.

### 2.3 Classical solution methods for agents

We categorize classical solution methods into three classes. Search methods include: 1) AI search methods (Russell & Norvig, 2020), e.g., A* and genetic programming, 2) operations research methods (Bertsekas & Tsitsiklis, 1996; Powell, 2011; 2022), e.g., mathematical programming, and 3) optimal control methods (Bertsekas, 2019; 2022), e.g., model predictive control (MPC). We treat optimization and planning as search. These methods work with a perfect model, or assume so. A notable example is Deep Blue. Bertsekas (2022) discusses lessons from AlphaZero for optimal, model predictive, and adaptive control.

Data-driven learning methods (Sutton & Barto, 2018; Bertsekas, 2019; Powell, 2022) include: 1) model-free methods, like imitation learning and model-free RL, e.g., Deep Q-Network (Mnih et al., 2015), and 2) model-based methods, like model-based RL, e.g. Dreamer (Hafner et al., 2025; 2020).

In the integration of learning and search methods (Sutton & Barto, 2018; Bertsekas, 2019; Powell, 2022), if with a perfect model or a perfect verifier, we can generate infinite perfect training data, e.g. AlphaGo series, with applications in maths (See Section 7). Otherwise, if without a perfect model, we have to estimate a model and approximate a simulator. And, we have to deal with estimation errors and bridge the simulation to reality gap. MuZero (Schrittwieser et al., 2020) is an example.

In LLMs, supervised fine-tuning (SFT) and Direct Preference Optimization (DPO) (Rafailov et al., 2023) are behavioral cloning. Reinforcement learning from human feedback (RLHF) (Christiano et al., 2017; Ouyang et al., 2022) learns a reward model first, then optimizes the

```
1: //AgentZero: model-based, with a perfect world model
2: //Agentμ: model-based, with a learned world model
3: //Agent∞: model-free, without a world model, thus can
       not perform Stage 3: decision time planning
4: // Stage 1. representation learning
5: collect ground truth data, train with SSL
6: train with auxiliary tasks, e.g., with meta RL
7: repeat
8:     // Stage 2. learning to make decisions
9:     // solve agent problems, e.g., with meta RL
10:    collect experience
11:    for each iteration do
12:        decision time planning, e.g., with MCTS
13:        update value function and policy
14:        Agentμ: also update/improve world model
15:    end for
16: until threshold met, e.g., for compute budget
17: // Stage 3. decision time planning (not for Agent∞)
18: meta-RL and/or MCTS
```

Algorithm 1: AgentZero / Agentμ / Agent∞

policy, thus it is inverse RL. SFT, DPO and RLHF are imitation learning. LLMs start to explore the integration of learning and search methods, in particular, OpenAI o1, DeepSeek R1 and their successors, in post-training and test time compute stages.

## 3 LLM agents in the era of experience inspired by games AI

### 3.1 AgentZero: AlphaZero-like LLM agents

We present AgentZero, an AlphaZero-like framework for LLM agents. We present its pseudo code in Algorithm 1, and compare it with current LLM agents in Table 1. Algorithm 1 and Table 1 are also for Agentμ and Agent∞, as discussed later.

AgentZero has three stages. In Stage 1 for representation learning (Bommasani et al., 2022), we employ self-supervised learning (SSL), e.g., GPT as in normal LLMs, to facilitate adaptation for further training, in particular, for RL training. Furthermore, for decision making problems, we may also train with auxiliary tasks to improve the representation (Jaderberg et al., 2017; Mirowski et al., 2017). Take program synthesis

| | current LLM agents | AgentZero / Agent$\mu$ / Agent$\infty$ |
|---|---|---|
| domain knowledge | agnostic to domain knowledge | leverage domain knowledge as much as possible, w.r.t. data collection, architecture design and algorithm design |
| ground truth | training data are usually not ground truth, esp. in the strong sense | ground truth data are considered by design; rich data types, esp. those in strong sense |
| iterative improvements | SSL during pre-training; infrequent then, e.g., in months | frequent by design, pre- & post-training, with algorithms like RL; update value, policy, and world model for Agent$\mu$ |
| base LLM | generalist, fixed | generalist or specialist, iterative improvements |
| scale of base LLM | large | large or small (around 7B or smaller) |
| world model | approximate, implicitly learned, infrequent update | AgentZero: perfect world model; Agent$\mu$: learned world model, frequent updates; Agent$\infty$, no world model |
| correctness | a common issue | a goal by design; aim to eliminate hallucinations |
| optimality | base LLM (SSL with next token prediction) and agent usually do not share the same goal | aim to achieve optimal agency by design; the goal may be the same during pre- and post-training stages |

Table 1: Comparison of current LLM agents vs AgentZero / Agent$\mu$ / Agent$\infty$. Note, GPT with next token prediction is agnostic to domain knowledge, although data do contain domain knowledge. For example, as we will discuss in Section 7.3, when a code LLM is trained using GPT with next token prediction, it does not leverage perfect code domain knowledge, like abstract syntax tree (AST), data flow graph (DFG), control flow graph (CFG), and the hierarchy of word, statement, function, class, and project in code. When we say "current LLM agents", we mean "current popular LLM agents", in particular, those based on fixed LLMs, with manually designed prompts and workflows.

as an example, there may be many auxiliary tasks, e.g., Tipirneni et al. (2024) consider abstract syntax tree (AST) path prediction and dataflow prediction.

In Stage 2 for learning to make decisions, we formulate agent (usually RL) problems, by defining corresponding state, action, and reward, as well as transition, value function, policy, etc. We collect experience in the form of (state, action, reward, next state). As for a normal RL problem, we take an action, observe a reward, and transition to the next state. We then update the value function and the policy. We may employ meta RL for better generalization (Finn et al., 2017; Wang et al., 2017; Hospedales et al., 2022; Beck et al., 2025). To make stronger decisions, we leverage decision time planning, e.g., with MCTS, which, as a results, generates better data. In Stage 3 for decision time planning, we may employ meta RL and/or MCTS, to make a stronger decision.

## 3.2 Agent$\mu$: MuZero-like AI agents

AlphaGo series are model-based RL with a perfect model. MuZero is model-based RL with a learned model. Inspired by MuZero, we propose Agent$\mu$, a MuZero-like framework for LLM agents. Agent$\mu$ is similar to AgentZero. The major difference is that Agent$\mu$ learns the world model, while AgentZero works with a perfect world model. AgentZero applies to problems with perfect rules, laws, and/or theorems, like many games, maths and coding tasks. Agent$\mu$ applies to problems starting with an approximate or no model/simulator. The significant difference between Agent$\mu$ and most current LLM agents hinges on iterative improvements based on ground truth of the base LLM and the world model.

Agent$\mu$ may also benefit from the progress in world model learning, e.g., Dreamer (Hafner et al., 2025; 2020), and reward model learning, e.g., RLHF (Christiano et al., 2017; Ouyang et al., 2022), and also earlier work about cooperative inverse RL (Hadfield-Menell et al., 2016) on alignment. We may need to consider reward

misspecification and reward hacking, e.g., inverse reward design (Hadfield-Menell et al., 2017). Imitation learning from human data is not enough to achieve superhuman performance (Silver & Sutton, 2025), as shown e.g. in AlphaGo Zero (Silver et al., 2017). We may design iterative versions of Dreamer and RLHF to make improvements of world or reward models based on experience from interactions with the world.

Note it is still infeasible to build fully autonomous and optimal LLM agents without human data. This is true even for maths and coding with formal systems (See Section 7). In MuZero, an agent collects ground truth data by interacting with the environment, the perfect game engine built from perfect game rules. We need to pay attention if the data collected by Agent$\mu$ are ground truth, and interactions with environments may be costly, e.g., for human-centric problems like education and physical problems like robotics. However, we may borrow ideas from games AI, and strive to make agents as autonomous and optimal as possible. Also note that even with a perfect world model, finding an optimal policy still requires exploration of algorithms. For example, Go is an ancient game with a perfect rule. However, only recently, AlphaGo found a superhuman strategy, decades after the invention of computers.

### 3.3 Agent∞: model-free agents

We also introduce the model-free Agent∞. It collects data by interacting with the environment, either offline or online. It does not attempt to learn a world model. An offline version is close to the current LLM agents, with enhancements from iterative improvements based on ground truth. Note, without a model, a model-free Agent∞ can not use decision time planning.

There are lots of RL/AI techniques to explore and exploit for better efficiency, generalization and scalability for AgentZero, Agent$\mu$ and Agent∞, such as, search space reduction with exploration, e.g., intrinsic motivation (Poesia et al., 2024; Barto, 2013), epistemic neural networks (Osband et al., 2023), and a survey (Amin et al., 2021), better representation with auxiliary tasks, e.g., Jaderberg et al. (2017) and Mirowski et al. (2017), causal representation, e.g., Schölkopf et al. (2021), better generalization with meta RL, e.g., Xiang et al. (2025), Setlur et al. (2025), Qu et al. (2025), Vettoruzzo et al. (2024), Beck et al. (2025), and Kirk et al. (2023), planning like MCTS, e.g., Zhang et al. (2023) and Guez et al. (2018), better abstraction with hierarchical RL, e.g., the option framework (Sutton et al., 1999) and Pateria et al. (2021), offline RL with batch data, e.g., Snell et al. (2023) and Levine et al. (2020), and better generalization with general value functions, e.g., Sutton et al. (2011) and Schaul et al. (2015).

By leveraging perfect domain knowledge, w.r.t. data collection, architecture design and algorithm design, we expect AgentZero, Agent$\mu$ and Agent∞ to improve interpretability, trustworthiness and safety innately. Hallucinations are inherent for current LLMs and LLM agents. We may mitigate or even eliminate hallucinations with domain knowledge during training and/or test stages, e.g., by parsing them out with game rules (Schultz et al., 2024), mathematical theorems (Trinh et al., 2024) and/or code syntax & semantics (Ugare et al., 2024). This may also improve efficiency and cost saving. With a stronger LLM, test time compute can be more efficient.

## 4 RQ1: Should we leverage domain knowledge?

Perfect domain knowledge implies strong prior information or strong inductive bias, thus leading to more efficient and effective learning systems. It is thus desirable to leverage such information in contrast to an AI agnostic to, or not leveraging enough, domain knowledge. Examples abound, like AlphaZero (Silver et al., 2018), AlphaGeometry (Trinh et al., 2024), and AlphaFold (Jumper et al., 2021; Abramson et al., 2024). Logic, maths and theoretical computer science are based on axioms and deductive reasoning, so that their domain knowledge is perfect. See Section 7.3 for discussion about coding. Domain knowledge in natural science and engineering represent the best knowledge humans have and may be under evolution, e.g., the understanding of physics evolves from animism, theism, Newtonian mechanics, to quantum mechanics (Silver & Sutton, 2025). In practice and in a relatively short term, we may incorporate the best domain knowledge to build an efficient AI system, e.g., for robotics, chip design, protein folding, drug design and chemical process. In theory and in a long term, learning from experience interacting with the world is an approach to achieve an evolution of domain knowledge (Silver & Sutton, 2025).

| category | world model | data | examples |
|---|---|---|---|
| AIZero | perfect world model | infinite perfect data from world model | AgentZero; AlphaZero, AlphaProof |
| AI$\mu$ | approximate world model | from approximate world model and real world | Agent$\mu$ (with learned, approximate models, yet frequent iterative improvements); robotics simulators; LLM-based methods, e.g., LLM agents (with implicit, approximate simulators) |
| AI$\infty$ | no world model | from real world | Agent$\infty$; AlexNet, LLMs, LLM agents |

Table 2: A taxonomy of AI models, with non-agentic and agentic examples, like AgentZero, Agent$\mu$, and Agent$\infty$ for AIZero, AI$\mu$, and AI$\infty$, respectively. $\mu \in (0, \infty)$ indicates the degree of model inaccuracy, which is related to the degree of effort to collect data or to bridge simulation to reality gap. Note, we put "LLM agents" as an example for both AI$\mu$ and AI$\infty$, depending on we treat them as model-based or model-free.

It is worthwhile to contrast domain knowledge with human knowledge, which may be heuristic to the best domain knowledge humans' have. How much human knowledge should be incorporated into an AI system is debatable. See, e.g., The Bitter Lesson (Sutton, 2019), A Better Lesson (Brooks, 2019), and Engineering AI (Kaelbling, 2019). Sutton (2019) advocates a meta-method to incorporate human knowledge. However, it appears that the discussion in Sutton (2019) does not include the case for perfect domain knowledge. This leaves a question: Should we learn such domain knowledge, e.g., axioms and those derived from them? We argue, similar to Brooks (2019) and Kaelbling (2019), but for perfect domain knowledge, that it is desirable to incorporate it in data collection, architecture design and algorithm design to achieve more efficient and effective learning. We tackle **RQ1.1: How to leverage domain knowledge?** when we discuss data collection, architecture design and algorithm design, and present a brief summary in Section 8.

We should differentiate the research questions about "Should we" and "How to" leverage domain knowledge. We should leverage domain knowledge as much as possible to exploit the strong and perfect prior information. However, how to leverage domain knowledge is problem dependent, e.g., it is relatively easier for games, maths and coding, and it may be non-trivial for problems with human-in-the-loop.

## 5 RQ2: How to attain ground truth data?

In the Introduction, we discuss what is ground truth, which also include the sources. A relevant question is: **Are synthetic data helpful?** Yes, if from a perfect world model or verifier. Yes, to some extent, if from a high fidelity model or simulator. Otherwise, something may go wrong. We illustrate this with the following categories of AI models: AIZero, AI$\mu$, and AI$\infty$, as shown in Table 2.

When there is a perfect model, we can build a perfect simulator to generate infinite training data in a digital world with correctness guarantee, much less costly than collecting data in a physical world. This includes cases with 1) a perfect rule, like AlphaZero, 2) a perfect verifier, like AlphaProof, and 3) a perfect dynamics, like a partial differential equation. We classify them as AIZero. AIZero can attain ground truth in the strong sense. There is a good chance to handle such cases in a way similar to AlphaZero. Note, compiling, passing tests, and execution results in coding, and numerical results for maths word problems, are ground truth in the broad sense, i.e., can't guarantee correctness without formal verification.

Many deep learning, or big data methods, like AlexNet, rely on a huge amount of annotated data. Model-free RL interacts with the environment online or offline to collect a huge amount of training data. Humans may make mistakes. Even well-calibrated datasets like ImageNet have label errors (Northcutt et al., 2021). For human-centric problems, e.g., education, we may have to rely on humans' feedback. So such ground truth data in the broad sense may be the best we can obtain. We classify these cases as AI$\infty$.

There are ways to help improve data efficiency, e.g., self-supervised learning, simulation, and model-based methods, and for RL in particular, intrinsic motivation and auxiliary signals. We call such approaches as AI$\mu$, where $\mu$ is a number between 0 and $\infty$, indicating the degree of inaccuracy of the model involved, or the degree of effort for manual data collection, or the degree of effort for interacting with physical environments.

AI$\mu$ approaches AIZero as the underlying model approaches perfect. AI$\infty$ implies there is no model involved. Admittedly, $\mu$ is only loosely defined.

High-fidelity simulators require significant efforts and are usually problem-specific. Data generated from such approximate models are close to ground truth in the broad sense. Such data may be helpful as training data. However, we need to bridge the simulation to reality gap. See, e.g., learning quadrupedal robot locomotion (Lee et al., 2020) and magnetic control of tokamak plasmas (Degrave et al., 2022).

Most LLMs are trained with Internet data, and RLHF involves human feedback, thus most LLMs are in AI$\infty$. LLMs may be a resourceful source for general information. There may be evidences that synthetic data are "superior to" data collected by humans. However, there may be issues like the evaluation method and the accuracy of the synthetic data. LLMs can serve as approximate but not perfect world models or simulators (Vafa et al., 2024; Wang et al., 2024c; Li et al., 2025). LLMs by themselves can not provide formal verification (Li et al., 2024; Yang et al., 2024a) or program synthesis with correctness guarantee (Olausson et al., 2024). A verifier is needed to filter out incorrect data, e.g., the LLM-modulo framework (Kambhampati et al., 2024). Knowledge graphs (KGs) or retrieval augmented generation (RAG) may help to mitigate inaccuracy or hallucinations from LLMs, however, still with limitations, e.g., Wu et al. (2025). When a model is trained with synthetic data sampled from LLMs, 1) if the data are filtered by a perfect verifier, then the model becomes AIZero, e.g., AlphaGeometry (Trinh et al., 2024); 2) otherwise, it is AI$\mu$.

Attempts of self-play with AI$\mu$ and AI$\infty$ may cause problems. See discussions about model collapse (Shumailov et al., 2024; Dohmatob et al., 2025). Stroebl et al. (2024) show the probability of false positives so that incorrect solutions may pass imperfect verifiers. Admittedly, a weak model may improve with data generated from a strong model, yet being limited by the strong model. LLMs may help RL (Murphy, 2024) with input pre-processing, e.g., autoformalization (Li et al., 2024; Yang et al., 2024a), prior (Yan et al., 2024), reward (Ma et al., 2024), world model (Yang et al., 2024b; Tang et al., 2024), and policy (Yao et al., 2023; Wang et al., 2024b). However, fixed, imperfect LLMs do not provide ground truth. The Bitter Lesson (Sutton, 2019) argues against reliance on human prior. Silver & Sutton (2025) treat LLMs as results from the era of human data, and discuss its limitations. For human-centric problems, human data are irreplaceable, however, ideally, with top experts' data. The Internet data are from a mixture of diverse expertises.

How to attain ground truth data? A brief summary follows. For problems like games, maths and coding, with a perfect world model or a perfect verifier, it is (relatively) easy to attain ground truth data in the strong sense (AIZero). However, ground truth data may be rare for many problems, esp. in the strong sense. Approximate world models or simulators, including LLMs, may help, but we need to handle the simulation to reality gap (AI$\mu$). For many problems, e.g., human-centric problems like education and physical problems like robotics, real world data are likely irreplaceable (AI$\infty$). Silver & Sutton (2025) outline a promising future with learning from experience interacting with the world, including a virtual, a physical, or a human world. However, there are still questions to tackle. How to deal with limited ground truth data? How to collect experience efficiently? How to design efficient learning algorithms, esp. with limited experience? We leave them as open problems.

## 6 RQ3: How to make iterative improvements?

### RQ3.1: Why are iterative improvements important?

We treat iterative improvement as a universal framework, with wide applications including gradient descent in optimization, boosting, expectation-maximum and temporal difference learning in AI, policy iteration in dynamic programming, close-loop feedback control, trial and error in animal learning, evolutionary methods like genetic programming, (agile) software development, and, evolution of our humankind. Sound iterative improvements require ground truth data. Iterative improvements are a natural approach for learning from experience by interacting with the world (Silver & Sutton, 2025). Interaction and embodiment are critical to agency (Bisk et al., 2020). Consider an example like learning bicycling. Books and videos may help. However, bicycling skills has to be achieved by practicing and learning from trial and error in the real world. Interactions and embodiments with the world are irreplaceable. Silver & Sutton (2025) emphasize the importance of experience and iterative improvements, and discuss the limitations of current LLMs and

human knowledge. Silver & Sutton (2025) present an example of the progress of humans' understanding of physics, from animism, theism, Newtonian mechanics, to quantum mechanics, and highlight the essence of the feedback loop with the real world: from hypotheses, experiments, results, to refinements of physical principles.

### RQ3.2: What are limitations of current LLMs?

Generative pre-trained transformer (GPT) with next token prediction is the most popular approach to LLMs, with Transformers (Vaswani et al., 2017) as the backbone for most LLMs. However, Transformers come with limitations. See, e.g., Dziri et al. (2023) for lack of compositionality, Deletang et al. (2023) for issues with generalization on non-regular tasks, and Merrill et al. (2024) for issues with state tracking problems like chess and code. There are studies for issues with popular approaches, e.g., Stechly et al. (2024b) for chain of thought (CoT) (Wei et al., 2022) in planning, Verma et al. (2024) for ReAct (Yao et al., 2023) in LLM agents, Stechly et al. (2024a) for self-verification in reasoning and planning, and Olausson et al. (2024) for self-repair for code generation.

Stoica et al. (2024) discuss the ambiguity of specifications in LLMs, and as a result, the challenges to achieve verifiability, debuggability, modularity, resuability, and automated decision making, and to make LLMs robust systems following engineering deciplines. A formal specification is the input to a formal verification, which guarantees the correctness of a program. Formal verification for deep learning or deep RL software systems is an emerging yet nascent research area, see e.g., Marabou 2.0 (Wu et al., 2024), $\alpha, \beta$-CROWN (Zhang et al., 2022), and NNV 2.0 (Lopez et al., 2023), König et al. (2024) and Landers & Doryab (2023).

There are alternative neural network architectures, e.g., state space models, like Mamba (Gu & Dao, 2024) and xLSTM (Beck et al., 2024), as well as TreeLSTM (Tai et al., 2015) and graph neural networks (GNN) (Corso et al., 2024; Wu et al., 2023). Wang et al. (2023) apply discontinuous networks for mathematics (Della Santa & Pieraccini, 2023) to contract design. Language models and multimodality models may use different Transformer architectures, e.g., Visual Transformers (ViT) for image recognition (Dosovitskiy et al., 2021). Schultz et al. (2024) study multi-action-value Transformer model for board games.

György et al. (2025) propose exact learning for deductive reasoning, which requires correctness on all inputs, rather than optimizing statistical performance w.r.t. a distribution. This is in stark contrast to the current statistical learning paradigm for LLMs, and may demand novel designs for data collection, (neural) architecture and algorithm.

A complex agent's representation may need more study. There are works for different levels of abstraction, other than token. See e.g., CodeBPE (Chirkova & Troshin, 2023) about tokenization for code, Pagnoni et al. (2024) about byte latent Transformer, Gloeckle et al. (2024) about multi-token prediction, The LCM team et al. (2024) about a sentence level representation, and Pertsch et al. (2025) for action tokenization for vision-language-action models.

RL enjoys a renaissance, esp. after OpenAI o1, for post-training, e.g., RLVR (Lambert et al., 2024), Xiang et al. (2025), Setlur et al. (2025), DeepSeek-R1 (DeepSeek-AI, 2025) and Kimi-k1.5 (Kimi Team, 2025). Ground truth has not attracted much attention, esp. in the strong sense. Also, there are few studies applying RL at the pre-training stage. The current LLMs, and the human data they are based on, lack of a feedback loop with the real world (Silver & Sutton, 2025; Bisk et al., 2020). To this end, we present a perspective paper with a pseudo code as a relatively concrete plan to overcome these issues.

### RQ3.3: Will scaling up current LLMs achieve reasoning and planning capacity to build agents?

"Scaling laws" (Kaplan et al., 2020; Hoffmann et al., 2022) refer to that the more data, the larger models or neural networks and the more compute, the better performance. Being true to some extent, some factors like high quality data, alternative neural network architectures and algorithms may make learning more efficient. It is also arguable that simply scaling up can lead to causation and general intelligence. "Compression is intelligence" may be true in the limit at convergence (Solomonoff, 1964; Ma et al., 2022). In practice, we may not have enough data, esp. for superhuman intelligence. Consider, e.g., a human-centric problem like education, without an objective objective / reward function, without a perfect world model or a perfect verifier, what is the definition of super-human and how to collect such "super-human" data? Ultimately,

we may have to rely on both data and model (Sutton, 2022), integrating empiricism and rationalism, to climb the ladder of causation from association to intervention to counterfactual (Pearl & Mackenzie, 2018). There are counter-examples to the scaling laws, e.g., a recent empirical study shows that larger and more instructable LLMs may have become less verifiable (Zhou et al., 2024).

The Bitter Lesson (Sutton, 2019) is often referred to as a support of scaling laws. However, the author, Richard Sutton is actually very critical on LLMs (Sutton et al., 2024). The Bitter Lesson highlights not only that general methods of learning and search can leverage computation, but also the importance of meta-methods. It is desirable to see the community start to explore search and meta-methods not only in post- but also in pre-training. However, ground truth, esp. in the strong sense, is still an issue. Following Silver & Sutton (2025), it is interesting to study how to scale up learning from experience.

**RQ3.4: Large vs small models? RQ3.5: Modularity? Generalist vs specialist?**

It is likely prohibitive to iteratively improve a monolithic, gigantic model. It is desirable to explore parameter efficient fine-tuning methods, e.g., Low-Rank Adaptation (LoRA) (Hu et al., 2022), and / or "small" models. Small language models (SLMs) ($\leq$ 7B) come with advantages w.r.t. cost-effectiveness, low inference latency, efficient development, easy customization, and easy adaptability (Wang et al., 2024a). Subramanian et al. (2025) show that SLMs can achieve comparable or better performance than much larger LLMs, and highlight the importance of data quality and specialty. Belcak et al. (2025) argue that SLMs are powerful, more suitable, and more economical for agents.

AlphaZero (47M), AlphaGeometry (125M), and AlphaFold (200M) outperform LLMs w.r.t. their specialities. In fact, the Python programming language and a calculator are not AI models, and are tiny in size comparing with LLMs, but can solve arithmetic problems perfectly, while all LLMs make mistakes. It is thus desirable to investigate specialized SLMs, exploit domain knowledge, explore pre-trainng, fine-tuning (last layers) and prompt techniques, computation efficient architectures, and model compression methods, including pruning, distillation, and quantization, to trade knowledge, data, architectures and algorithms for scale and compute.

Stoica et al. (2024) discuss that modularity is critical for building reliable systems, with examples from computer, software, automotive, and construction industries. There are efforts for modular design for LLMs. Zaharia et al. (2024) discuss the shift from models to compound AI systems. Ostapenko et al. (2024) propose to build and reuse a libraries of LoRAs for modular LLMs. Yadav et al. (2024) survey model MoErging for collaborative learning. Mixture of Experts (MoE) or routing (Pfeiffer et al., 2023), e.g., Transformer$^2$ (Sun et al., 2025), essentially implement multiple models with a single neural network.

To make a model, esp. small ones, perform well, we may need to focus on specific task(s), rather than to aim for generality. Intuitively, a general intelligence system strikes the balance among multiple objectives and constraints. E.g., Wolf et al. (2024) discusses that making capabilities more distinct might improve their reliability but at the cost of requiring more specific prompts and as the number of objectives increases, maintaining consistent performance across all of them becomes exponentially harder. The evolution from ENIAC to iPhone indicates that further progress in AI calls for more innovations. Complex tasks can be achieved by collaboration of multiple models. We expect specialized, smaller models to play important roles.

We basically treat modularity as specialist models, in stark contrast to the current popular approach that LLM agents are based on a monolithic LLM. It is already a big topic and may cause lots of discussions or even debates, so we do not intend to discuss finer-granularity of modularity for a single specific agent, e.g., there would be lots of sub-modules for a coding agent, e.g., specification, generation, compiling, testing, verification, security, safety, UI/UX, etc.

**RQ3.6: What algorithms are suitable for iterative improvements?**

RL is a natural framework for an agent to interact with an environment, receives (ground truth) feedbacks and makes improvements iteratively. Neuro-symbolic methods may overcome the limitations of LLMs, or neural methods, e.g., in solving arithmetic problems. Evolutionary methods are iterative by nature and are both competitive and collaborative with RL, e.g., FunSearch (Romera-Paredes et al., 2024) and AlphaEvolve (Novikov et al., 2025). We expect RL, evolutionary and neuro-symbolic methods, among other AI methods, to play important roles in building agents. See Sections 3 and 7 for more detail. We quote from David

Silver's recent talk (Silver, 2024): "LLMs alone are insufficient to achieve superhuman intelligence." "Super-scaled RL provides a clear pathway to superhuman intelligence." "Perfect verifiability allows us to generate provable correct proofs." Silver & Sutton (2025) draw a blueprint for achieving super-human intelligence with experience and reinforcement learning. We provide a more pragmatic and concrete plan.

## 7 Case studies

### 7.1 Games agents

For many games, like chess, go, Atari and Starcraft, deep RL has made significant, usually superhuman achievements. GPT may not be an efficient and effective approach. However, investigating games with GPT may shed light on LLMs' reasoning and planning capacity (Ruoss et al., 2024).

Schultz et al. (2024) pre-train a Transformer, the multi-action-value (MAV) model, on textual game data, to function as a world model, a value function and a policy function for several board games with perfect information. A world model requires state tracking, legal move prediction and terminal state detection. Schultz et al. (2024) employ external and internal planning and achieve Grandmaster-level performance in chess with language models of 2.7B parameters, however, not the competence of AlphaZero or MuZero yet. Schultz et al. (2024) is Agent$\mu$, following MuZero.

For games with verbal communication like the seven-player game of Diplomacy, agents powered by LLMs would be a great fit. Cicero (Bakhtin et al., 2022) integrates an LLM with planning and RL algorithms in Diplomacy to infer players' beliefs and intentions from conversations and to generate dialogues for negotiation and tactical coordination. Human players' experience is critical, and it is a mixture of Agent$\mu$ and Agent$\infty$.

### 7.2 Maths agents

AlphaGo series inspire AlphaTensor (Fawzi et al., 2022) for discovering faster matrix multiplication algorithms, AlphaDev (Mankowitz et al., 2023) for discovering faster sorting algorithms and AlphaProof (AlphaProof and AlphaGeometry teams, 2024) for proving mathematical statements in a formal language. AlphaProof has a formalizer network and a solver network. The formalizer network is obtained by fine-tuning a Gemini model. It translates problems from natural language to formal statements. The solver network leverages the AlphaZero algorithm, searching for proofs or disproofs of the problems in the formal language Lean. Then verified formal proofs become training data for the next iteration.

### 7.3 Coding agents

For both maths and coding, there is precise, rich and valuable domain knowledge. It is thus desirable to investigate the best designs of data collection, architecture and algorithms. Treating programming languages as a natural language using GPT with next token prediction wastes such valuable information. A code model may operate at multiple levels of abstraction, from word, statement, function, class, to project, which calls for more studies. See Section 6 for more discussion about alternative architectures.

There are papers exploiting code domain knowledge for LLMs, like Abstract Syntax Tree (AST), control flow graphs (CFG), data flow graphs (DFG), and compiler intermediate representation, with trees and/or graphs, e.g., StructCoder (Tipirneni et al., 2024), FAIR (Niu et al., 2024), GrammarT5 (Zhu et al., 2024), and PPOCoder Shojaee et al. (2023). Bounsi et al. (2024) propose the TransNAR architecture, combining Transformers and graph neural network (GNN), to achieve robustness for algorithmic tasks. RL is studied in code LLMs, e.g., CodeRL (Le et al., 2022), PPOCoder (Shojaee et al., 2023), RLSF (Jha et al., 2024), RLEF (Gehring et al., 2025), PGTD (Zhang et al., 2023), and DeepSeek-Coder-V2 (DeepSeek-AI et al., 2024), as well as general LLMs, e.g., Tulu 3 (Lambert et al., 2024).

Maths and coding, with a formal verifier, come close to games AI, with a perfect game engine / simulator. However, the goal of fully autonomous and correct maths or coding agents is likely infeasible, due to the impossibility results, e.g., Gödel's incompleteness theorems and the halting problem. Hüttel (2025) discuss program synthesis and LLMs. Brooker & Desai (2025) discuss a hybrid approach with formal verification

and semi-formal methods like testing and fuzzing. This being said, there are and will be lots of academic and business opportunities and we may try our best, by iterative improvements based on ground truth. Before becoming fully autonomous, a maths or a coding agent is a mixture of 1) Agent$\mu$, for feedback from a verifier for both maths and coding and feedback from compiler, tests, etc., and 2) Agent$\infty$ for humans' feedback.

### 7.4 Agents with approximate simulators

This is about wide applications in natural science, engineering, economics, finance, social science, or any disciplines, where we may have scientific or engineering principles, but still need to rely on high fidelity yet approximate simulators. It is critical to bridge the simulation to reality gap, see e.g., Wagenmaker et al. (2024). Robotics is a representative application in this category (Ibarz et al., 2021). Lavin et al. (2022) discuss the merge of scientific computing, scientific simulation, and AI. LLMs can be treated as approximate simulators (Vafa et al., 2024; Wang et al., 2024c; Li et al., 2025). Thus most current LLM agents fall into this category, however, without iterative improvements of underlying LLMs, and maybe also without ground truth, as in Agent$\mu$. Popular approaches like prompt engineering, Chain of Thought, ReAct, and LLM multi-agent, as well as the recent test time compute, can be regarded as attempts to bridge the simulation to reality gap.

### 7.5 Agents with facts

We refer this type of agents to those that depend on factual information, e.g., in the form of knowledge graph (KG) or retrieval augmented generation (RAG) (Lewis et al., 2020; Gutierrez et al., 2024; Wu et al., 2025). This may be complementary to other types of agents. We note that KGs and RAGs are usually not perfect. How to make them perfect or bridge the factuality gap appears as an open problem.

### 7.6 Agents with human-in-the-loop

Here we refer to agents with humans playing a major role, e.g., in education and healthcare. For many NLP problems, e.g., translation and summarization, objectives are subjective, and performance metrics like BLEU, ROUGE and perplexity are heuristic. This may also apply to many human-centric problems in social sciences like psychology, cognitive science, and behavioural science. Even there may be principles, they are not in the perfect sense, and behavioural study with humans are essential.

As a result, we may not be able to define a precise objective or reward function for the decision problem. RLHF is a principled way to learn a reward function, however, with human preference data. See, e.g., Casper et al. (2023) for RLHF and Retzlaff et al. (2024) for human-in-the-loop RL. In such cases, a key issue is how to guarantee the information is ground truth. Interactions with humans esp. experts are the best effort, yet may be the best data we can obtain. See e.g. Chakrabarty et al. (2025) about writing quality. We propose to make iterative improvements based on ground truth following Silver & Sutton (2025).

We note that for maths, coding, scientific, engineering and factual agents, before becoming fully autonomous, human-in-the-loop is inevitable. So the discussions here may complement to those agents to some extent.

## 8 Compare with current approaches

In the above, we discuss RQ1.1: How to leverage domain knowledge? when we discuss data collection, architecture design and algorithm design. Here is a brief summary. In Algorithm 1, collecting ground truth data, designing auxiliary tasks, and even collecting experience, may require domain knowledge. In Section 5 for RQ2: How to attain ground truth data?, we also discuss the question: Are synthetic data helpful?. Domain knowledge is essential for data collection, in particular, for a perfect world model (e.g., game rules), for a perfect verifier (e.g., maths and coding), or for a high-fidelity simulator (e.g., physical laws underlying robotics simulators). When we discuss RQ3.2: What are limitations of current LLMs?, we mention alternative neural network architectures, e.g., TreeLSTM, GNN, discontinuous networks for mathematics (Della Santa & Pieraccini, 2023), contract design (Wang et al., 2023), Visual Transformers (ViT) for image recognition (Dosovitskiy et al., 2021), multi-action-value Transformer model for board games

(Schultz et al., 2024), and different levels of abstraction, e.g., CodeBPE tokenization for code (Chirkova & Troshin, 2023). All these leverage domain knowledge. The architecture may influence algorithms. In Sections 7.1 for games agents, 7.2 for maths agents, and 7.3 for coding agents, we present concrete examples for how to leverage domain knowledge, in particular, for coding, the precise domain knowledge of Abstract Syntax Tree (AST), control flow graphs (CFG), data flow graphs (DFG), and compiler intermediate representation. The hierarchy of word, statement, function, class, and project in code makes a hierarchical attention mechanism, rather than next token prediction, very promising.

Most current approaches directly apply LLMs to downstream tasks, including agents, by employing auxiliary techniques like prompt engineering, CoT, ReAct, RAG, workflows, the Model Context Protocol (MCP), and/or test time compute, based on one or multiple fixed LLMs, without improving the underlying LLM(s). The significant difference with our approach is iterative improvements to the base LLM(s). Ground truth may also be a significant difference, when adjustments are based on information generated by LLM(s) without formal verification. "Foundation models" were supposed to learn good representations, and adapt to downstream applications (Bommasani et al., 2022). We propose to revisit this view and highlight the importance of adaptation of the LLMs themselves, with iterative improvements based on ground truth. See e.g., Cosmos (NVIDIA et al., 2025).

We propose to build specialist models, rather than generalist LLMs. We propose to divide and conquer, rather than following a holistic appraoch. See e.g., the integration of the Transformer as a long term memory with the Soar cognitive architecture (Laird et al., 2023). We highlight the importance of ground truth, and bridge the gap between approximate models to reality, in contrast to several current approaches based on fixed, imperfect LLMs: 1) synthetic data generated by LLMs; 2) self-play with LLMs; 3) treat LLMs as policy, value function, and world model; 4) rely on LLMs for decision making and agency. We highlight the importance of interactions and embodiment, and thus iterative improvements. See, e.g., Silver & Sutton (2025) and Bisk et al. (2020) for the importance of experience. We propose to explore and exploit domain knowledge, w.r.t. data collection, architecture design and algorithm design, leverage both learning and search, make iterative improvements based on ground truth. This is in contrast to 1) scaling up GPT with next token prediction to achieve general superhuman performance, and 2) compression is intelligence. We propose to build many specialized agents, with one or many coordinator agents, and then form the multi-agent system with specialists, making iterative improvements based on ground truth, in contrast to the current popular approach in which agents are based on general, fixed, imperfect LLMs.

Autonomy and optimality are two critical features for agents. Many current LLM agent works use the success rate as a performance metric, and use manual, heuristic workflows when constructing prompts. We propose to return to aiming at autonomy and optimality, as in Sutton & Barto (2018) and Russell & Norvig (2020). Both iterative improvements and ground truth are important ingredients for achieving such a goal.

In academia, there are discussions even debates about LLMs' reasoning and planning capacities, which are fundamental for building agents. In practice, after around two years' endeavours by many people worldwide, few successful agents emerge, esp. considering autonomy and optimality. We highlight the importance of iterative improvements and ground truth, along with perspectives on specialized and small models. We expect to help make progress in the research and commercialization of LLM agents.

By exploring and exploiting domain knowledge, w.r.t. data, architectures and algorithms, and following iterative improvements based on ground truth, we expect to improve interpretability (Barredo Arrieta et al., 2020; Zhao et al., 2024; Bereska & Gavves, 2024; Milani et al., 2024), trustworthiness (Huang et al., 2024; Liu et al., 2024) and safety ("davidad" Dalrymple et al., 2024; Gu et al., 2024) innately. This may address the issue of in-context reward hacking in feedback loops (Pan et al., 2025). This may also improve efficiency and cost saving, as advocated, e.g., in the Abstraction and Reasoning Corpus (ARC) Prize. Kapoor et al. (2024) discuss benchmarks for agents, highlighting jointly optimization of accuracy and cost.

Chen et al. (2025) shows that self-play fine-tuning improves LLMs. It works in the way of generative adversarial (GAN) (Goodfellow et al., 2020). A hypothesis is: The initial LLM is not trained well enough yet. Thus it is critical to make iterative improvements based on ground truth. Chu et al. (2025) study SFT vs. RL for memorization vs generalization for post-training. Rohatgi et al. (2025) show that GPT with next token prediction is imitation learning with performance barrier for error amplification. This may explain

why LLMs demonstrate reasoning and planning capacity, however, with issues: quality and coverage of data and efficiency and competency of learning. Moreover, imitation learning may not be enough to achieve super-human performance (Silver et al., 2017).

## 9 Discussion

Silver & Sutton (2025) draw a blueprint for (reinforcement) learning from experience. We provide a more pragmatic plan as a complement. We propose LLM agent frameworks in the era of experience inspired by the successes of games AI. We highlight the importance of both iterative improvements and ground truth, propose to explore and exploit domain knowledge w.r.t. data collection, architecture design and algorithm design, with decision time planning and meta RL at both pre- and post-training stages. We also present case studies. Iterative improvements based on ground truth helps an agent ground in the experience. Iterative improvements are incremental by nature, and may be offline/batch or online learning (Sutton & Barto, 2018; Laird et al., 2023). In practice, it can be more frequent than offline, yet not as frequent as online learning.

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
