# OpenReview forum: "Iterative Improvements Based on Ground Truth: Building LLM Agents in the Era of Experience Inspired by Games AI"
_TMLR — Rejected by TMLR_

### Review · Reviewer_GeGt · 2025-06-09

**Summary Of Contributions:**

This paper proposes a new perspective on developing successful LLM agents with emphasis on iterative improvements based on ground truth. The core ideas are drawn from the successes of Games AI, and the authors introduce three agent frameworks: AgentZero, Agent$\mu$, and Agent$\infty$, which are analogous to AlphaZero, MuZero, and model-free methods like DQN, respectively/

**Audience:**

Yes

**Claims And Evidence:**

No

**Requested Changes:**

- I would like to ask the authors to change the tones of the answers for the research questions. I personally feel that the answers for the research questions are too strong and even dubious. If the authors still continue to make such strong arguments, they need to provide more evidences. I agree with the authors that AlphaZero or AlphaFold enjoyed domain knowledge, but this does not lead to a clear conclusion that we should leverage domain knowledge in any applications.

**Strengths And Weaknesses:**

### Strengths

- This paper is well-written and organized. I found this paper is easy to follow.
- The topic of this paper is quite timely, so this paper will be paid much attention.
- The authors have extensively surveyed in the literature, from early works to the most recent papers.

### Weaknesses

- I feel that the arguments are sloppy in some places. For example, the authors mention that "The answer should be clear" for the RQ1: Should we leverage domain knowledge? However, I think it is not that easy to say that the answer is clear given the reward hacking or difficulty in getting precise, rich and valuable domain knowledge. The authors seem to be aware of this, but I would like to ask the authors to avoid making such overly strongly arguments.

- The paper offers a useful overview of existing work, but it reads more like a compilation of prior studies than a source of new insights for the reader. While this approach is entirely appropriate for a survey article, it may make the manuscript somewhat less engaging.

---

> ### Author Response · Authors · 2025-06-24
> **We appreciate the helpful review by Reviewer GeGt.**
>
> Please check the Official Comment on top.
>
> To respond to Reviewer GeGt‘s Requested Changes, as well as the listed Weaknesses for the tone, we will replace the following:
> “The answer should be clear: precise, rich and valuable domain knowledge implies strong prior information or strong inductive bias, thus leads to more efficient and effective learning systems. It is thus a waste of information if an AI is agnostic to, or does not leverage enough, domain knowledge. We treat it as a common sense in AI.”
> with
>
> “We argue that precise, rich and valuable domain knowledge implies strong prior information or strong inductive bias, thus leading to more efficient and effective learning systems. It is thus desirable to  leverage such information in contrast to an AI agnostic to, or not leveraging enough, domain knowledge.”
>
> We will also revise the tones in other places as suggested by Reviewer GeGt.
>
> We did contrast (perfect) domain knowledge with (heuristic) human knowledge:
> “It is worthwhile to contrast domain knowledge with human knowledge, which may be heuristic to the best domain knowledge humans’ have. How much human knowledge should be incorporated into an AI system is debatable. See, e.g., The Bitter Lesson (Sutton, 2019), A Better Lesson (Brooks, 2019), and Engineering AI (Kaelbling, 2019).”
> We plan to add one sentence:
>
> “Sutton (2019) advocates a meta-method to incorporate human knowledge. However, it appears that the discussion in Sutton (2019) does not include the case for perfect domain knowledge. This leaves a question:  Should we learn such domain knowledge, e.g., axioms and those derived from them? We argue, similar to Brooks (2019) and Kaelbling (2019), but for perfect domain knowledge, that it is desirable to incorporate it in data, architecture and algorithms to achieve more efficient and effective learning.”
>
>
> Weakness:
>
> “it is not that easy to say that the answer is clear given the reward hacking or difficulty in getting precise, rich and valuable domain knowledge.”
>
> It is true we need to be careful when there may be reward hacking or it is difficult to get perfect domain knowledge.
> When there may be reward hacking, we need to pay attention to the reward design, which is a research topic by itself, e.g., Inverse Reward Design, NIPS 2017 (NIPS renamed to NeurIPS in 2019).
>
> We argue to leverage domain knowledge as much as possible, esp. when the domain knowledge is perfect, like maths axioms and physical laws.
> This is in contrast to the popular approach to training LLMs, i.e., GPT with next token prediction, which is agnostic to domain knowledge.
>
> Weakness: overview of existing work
>
> Table 2 is our attempt to provide our analysis of many AI models with a taxonomy wrt world model and data, and with examples. We use Table 2 and AIZero/AI$\mu$/AI$\infty$ to discuss the issues with synthetic data, the importance of accurate world models, and the necessity of bridging the simulation to reality gap.

---

### Review · Reviewer_Jppn · 2025-06-11

**Summary Of Contributions:**

This paper is a perspective paper that proposes a series of LLM agent frameworks: AgentZero, Agent$\mu$, and Agent$\infty$. AgentZero aims to learn from a perfect world model with reliable verifiers or known rules; Agent$\mu$ learns both a world model and decision-making, and Agent$\infty$ learns in a model-free manner. Based on this group of frameworks, the paper discusses several design choices in the framework, including leveraging domain knowledge, source of ground truth data and how to make iterative improvements. The paper provides a few examples of LLM agents as case study.

**Audience:**

Yes

**Broader Impact Concerns:**

The paper does not include a broader impact statement. As this paper does not propose any concrete instance of new methods or conduct any experiment, I think the work has no particular ethical concern beyond general LLM works.

**Claims And Evidence:**

No

**Requested Changes:**

1. Address the problems in weaknesses.

2. Fix inconsistent formats. for example,  a subsection (6.1) is allocated for RQ3.1, but not for RQ3.2, 3.3, etc.

3. Make the difference between AgentZero, Agent$\mu$ and Agent$\infty$ in Alg. 1 clearer (e.g. using different colors); currently it is not clear to discriminate diffrent control flows for different algorithms.

**Strengths And Weaknesses:**

**Strengths**

1. the paper makes a wide investigation on a variety of LLM papers in this field, including GameAI and many different applications for LLM / LLM agents. I appreciate the ambition of this paper to build a unified model for LLM agents.

2. The high-level idea of this paper is generally well-conveyed through Tab. 1, Alg. 1 and discussions in research questions.

**Weaknesses**

1. The structure of this paper is unclear.

a) In Sec. 7 the authors mention "Schultz et al. (2024) is Agent$\mu$" (Sec. 7.1); "a math or coding agent is a mixture of Agent$\mu$ and Agent$\infty$" (Sec. 7.3), in which AgentZero/Agent$\mu$/Agent$\infty$ serves as a dichotomy for existing methods, which seems to use the name as a dichotomy for LLM agents; however, other parts of the paper (e.g. Sec. 8 and AIZero/AI$\mu$/AI$\infty$) suggests otherwise, that AgentZero/Agent$\mu$/Agent$\infty$ are standalone frameworks.

b) The authors claim to discuss RQ 1.1 when discussing data collection, architecture design and algorithm design, but there is only scattered discussion on architecture design (e.g. in RQ 3.2 and 3.3 with alternative architectures) which does not mention leveraging domain knowledge. Discussions on algorithm design, such as those in RQ 3.6 or Sec. 7.3, also does not mention leveraging domain knowledge.

c) Some parts of the paper does not seem closely connected. For example, how is modularity (RQ 3.5) connected to making iterative improvements?

2. The contribution of this paper is unclear. The authors made no attempt in building an instance with concrete choice of algorithm components on a particular question is provided in the paper despite discussing research questions in detail, nor do they run any experiment to prove the validity of their framework. Thus, the current claim for adopting the proposed framework is not convincing enough.

3. Some statement in this paper is unclear.

a) What is modularity for LLMs? The paper discusses briefly on modularity's importance in building LLM agents, but never defines modularity. According to the paper, the definition of modularity varies from multi-agent system ("compound AI systems") to architecture (MoE).

b) What does it mean by "self-supervised learning to improve the representation, e.g. with GPT as in normal LLMs" (Sec. 3.1)? GPT itself is not a self-supervised learning method and the representation in "normal LLMs" is not learned by a standalone GPT.

c) AgentZero/Agent$\mu$/Agent$\infty$ "aims to eliminate hallucinations" (Tab. 1), but the paper does not discuss anything about how this can be achieved (except only briefly mentioned once in RQ2 that knowledge graphs and RAG are imperfect solutions).

d) in RQ3.6, the authors mention that "We expect RL, evolutionary and neuro-symbolic methods, etc. to
play important roles in building agents." While I do not disagree with the idea, I feel the reasoning is a bit ad hoc; for example, why is evolutionary and neuro-symbolic promising? What are the other options, and why are they not feasible? These are all discussions that I feel missing from the paper.

4. Some statement in this paper is inaccurate or wrong.

a) The authors claim that current LLM agents are agnostic to domain knowledge, while the proposed framework leverage domain knowledge as much as possible. This is inaccurate - current state-of-the-art LLMs, such as o3, already stores a good amount of domain knowledge with its pre-training (o3 already performs well on graduate-level domain-specific questions (GPQA diamond, https://arxiv.org/abs/2311.12022), and currently LLM's performance is quickly rising on the most cutting-edge domain knowledge test (Humanity's Last Exam, https://arxiv.org/abs/2501.14249).

b) current LLM agents has large scale of base LLM (Tab. 1) - this is wrong. Many LLM agent papers (e.g. PAE https://arxiv.org/abs/2412.13194, VTool-R1 https://arxiv.org/pdf/2505.19255) work with 7B or even smaller models.

c) The authors mention that current LLM agents conduct iterative improvements during pre-training and is not frequently improved in months. However, isn't any LLM agent trained with RL without any other algorithm design an instance of Agent$\infty$, since RL is proposed by the authors in RQ 3.7 for iterative improvements?

---

> ### Author Response · Authors · 2025-06-24
> **We appreciate the critical review by Reviewer Jppn.**
>
> We will make the following points clearer in the revised version.
> One critical point is, we propose to pre-train a model with iterative improvements, ideally when enough resources are available. This has chances to build better base models, and makes fine-tuning and prompting more efficient and effective. Please check the Official Comment on top for more detail.
>
> Weakness 1. a) AgentZero/Agent$\mu$/Agent$\infty$ vs AIZero/AI$\mu$/AI$\infty$
>
> “We propose AgentZero and Agent$\mu$, agent frameworks with perfect and learned world models, following AlphaGo and MuZero, respectively. We also propose a model-free framework Agent$\infty$.”
>
> AIZero/AI$\mu$/AI$\infty$ are introduced in Section 5 RQ2: How to attain ground truth data? As Table 2 shows, there are two important features, world model and data, for AIZero/AI$\mu$/AI$\infty$, to differentiate AI models. These are complementary to AgentZero/Agent$\mu$/Agent$\infty$.
>
> Table 1 shows that we compare current LLM agents to  AgentZero/Agent$\mu$/Agent$\infty$ w.r.t. several factors.
>
> Weakness 1. b) RQ1.1: How to leverage domain knowledge?
>
> We choose the way to discuss RQ1.1 “when we discuss data collection, architecture design and algorithm design”, to attempt to make it follow a natural flow and to avoid repetition.
> Please check the Official Comment on top, for A brief summary for RQ1.1.
>
> Weakness 1. c) modularity vs iterative improvements
>
> “It is likely prohibitive to iteratively improve a monolithic, gigantic model.”
> We propose to (pre-)train modular, specialized, smaller models, so that it is feasible and more efficient to iteratively improve models and to achieve better and better models, more frequently.
> See also response to Weakeness 3 a).
>
> Weakness 2. Contributions & 3. a) modularity
>
> See Official Comment on top.
>
> Weakness 3. b) self-supervised learning vs the representation
>
> GPT with next token prediction or self-supervised learning is the popular approach to LLMs, and conduct representation learning, simply put, feature learning in traditional machine learning.
>
> Weakness 3. c) aims to eliminate hallucinations
>
> Section 3.3
> “We may mitigate or even eliminate hallucinations with domain knowledge during training and/or test stages, e.g., by parsing them out with game rules (Schultz et al., 2024), mathematical theorems (Trinh et al., 2024) and/or code syntax & semantics (Ugare et al., 2024). This may also improve efficiency and cost saving. With a stronger LLM, test time compute can be more efficient.”
>
> Weakness 3. d)  AI methods
>
> We agree that each AI method may play a role in building agents.  We hope to limit the submission to 12 pages. Neuro-symbolic methods may overcome the limitations of LLMs. The evolutionary method is iterative by nature.
>
> Weakness 4. a) current LLM agents are agnostic to domain knowledge
>
> We will make it clearer in the revision that when we say “current LLM agents”, we mean “current popular LLM agents”, e.g., those based on LLMs, with prompt engineering and manually designed workflows, as well as with fine-tuning or post training. When the LLMs were pre-trained  following the approach of GPT with next token prediction, they did not leverage domain knowledge.
>
> Also, although As Reviewer Jppn mentioned that “current state-of-the-art LLMs, such as o3, already stores a good amount of domain knowledge with its pre-training”, the stored knowledge may not be perfect, so that there are errors or hallucinations. And our proposal is to mitigate or even eliminate such errors by incorporating perfect domain knowledge w.r.t. data collection, architecture design and algorithm design, even in the pre-training stage. See also the response to Weakness 3. c) above.
>
> Weakness 4. b) current LLM agents has large scale of base LLM
>
> Again, when we say “current LLM agents”, we mean “current popular LLM agents”. See the response to Weakness 4. a)  for more detail.
>
> When there are smaller, successful models, they are actually the evidence to support our proposal.
>
> Thanks for pointing to paper. However, both PAE AND VTool-R1 are about fine-tuning with RL. We propose to pre-train with iterative improvements. Moreover, their feedback is based on LLMs, and thus not ground truth. See Section 5 RQ2: How to attain ground truth data?
>
> Weakness 4. c) iterative improvements with RL
>
> Most LLM agents are based on fixed, imperfect LLMs, with infrequent iterative improvements.
> The current hot direction is to use RL for post-training.
> There are rare papers about pre-training LLM agents with RL.
> One example is CodeRL for code generation through pretrained models and deep RL.
>
> After our submission, we saw this arxiv paper (June 9, 2025)
> Reinforcement Pre-Training, https://arxiv.org/abs/2506.08007
> However, it works with GPT next token prediction, for a general model, and does not leverage domain knowledge (enough). There still appears to be a big room, if our proposal is leading to a more efficient and effective direction.
>
> We will fix issues in Requested Changes.

---

> > ### Author Response · Authors · 2025-06-24
> > **Longer version rebuttal to Reviewer Jppn (part 1)**
> >
> > This is the longer version, and we recommend you read it.
> > We thought the maximum length of the whole rebuttal to a reviewer is 5000 characters, so we shortened it as above.
> >
> > As a general response, we will make the following points clearer in the revised version.
> > One critical point is, we propose to ideally pre-train a model with iterative improvements. This has chances to build better base models, and makes fine-tuning and prompting more efficient and effective. Please check the Official Comment on top for more detail.
> >
> > We address weaknesses one by one in the following.
> >
> > Weakness 1. a) AgentZero/Agent$\mu$/Agent$\infty$ vs AIZero/AI$\mu$/AI$\infty$
> > “We propose AgentZero and Agent$\mu$, agent frameworks with perfect and learned world models, following AlphaGo and MuZero, respectively. We also propose a model-free framework Agent$\infty$.”
> >
> > AIZero/AI$\mu$/AI$\infty$ are introduced in Section 5 RQ2: How to attain ground truth data? As Table 2 shows, there are two important features, world model and data, for AIZero/AI$\mu$/AI$\infty$, to differentiate AI models. These are complementary to AgentZero/Agent$\mu$/Agent$\infty$.
> >
> > Table 1 shows that we compare current LLM agents to  AgentZero/Agent$\mu$/Agent$\infty$ w.r.t. several factors.
> >
> >
> > Weakness 1. b) RQ1.1: How to leverage domain knowledge?
> >
> > We choose the way to discuss RQ1.1 “when we discuss data collection, architecture design and algorithm design”, to attempt to make it follow a natural flow and to avoid repetition.
> > We plan to add a (sub)section for a brief summary to answer RQ1.1. Please check the 2nd part of Official Comment on top, for A brief summary for RQ1.1: How to leverage domain knowledge?
> >
> > Weakness 1. c) how is modularity (RQ 3.5) connected to making iterative improvements?
> >
> > “It is likely prohibitive to iteratively improve a monolithic, gigantic model.”
> > We propose to (pre-)train modular, specialized, likely smaller models, so that it is feasible and more efficient to iteratively improve models and to achieve better and better models, more frequently than months, likely in weeks or days, or even more frequent.
> > See also the response to Weakeness 3 a) below.
> >
> > Weakness 2. The contribution of this paper is unclear.
> >
> > In Introduction: “We make the following contributions for building LLM agents: 1) We highlight the importance of both iterative improvements and ground truth. 2) We highlight the importance of domain knowledge, w.r.t. data collection, architecture design and algorithm design, and to strike a balance between a) scale and b) performance, efficiency and cost. 3) We expect to help improve the interpretability, trustworthiness and safety. 4) We briefly survey relevant works, make a critical examination, and propose a potential solution.”
> >
> > We also add an Official Comment on top.
> >
> > Weakness 3. a) What is modularity for LLMs?
> >
> > “We propose to build specialist models, rather than generalist LLMs. We propose to divide and conquer, rather than following a holistic approach.” in Section 8 Compare with current approaches.
> > We basically treat modularity as specialist models, in a stark contrast to the current popular approach that LLM agents are based on a monolithic LLM like GPT-4. It is already a big topic and may cause lots of discussions or even debates, so we do not intend to discuss finer-granularity of modularity for a single specific agent, e.g., there would be lots of sub-modules for a coding agent, e.g., specification, generation, compiling, testing, verification, security, safety, UI/UX, etc.
> >
> > Weakness 3. b) “self-supervised learning to improve the representation”
> >
> > GPT with next token prediction or self-supervised learning is the popular approach to LLMs, e.g.,  ChatGPT, Claude and Gemini, and conduct representation learning, in the sense of e.g. Representation Learning: A Review and New Perspectives, https://arxiv.org/pdf/1206.5538. Simply put, consider representation learning as feature learning in traditional machine learning.
> >
> >
> > Weakness 3. c) AgentZero/Agent$\mu$/Agent$\infty$ "aims to eliminate hallucinations"
> >
> > We have the following discussion under Section 3.3 Agent$\infty$ : model-free agents
> > “By leveraging domain knowledge, w.r.t. data, architectures and algorithms, we expect AgentZero, Agent$\mu$ and Agent$\infty$  to improve interpretability, trustworthiness and safety innately. Hallucinations are inherent for current LLMs and LLM agents. We may mitigate or even eliminate hallucinations with domain knowledge during training and/or test stages, e.g., by parsing them out with game rules (Schultz et al., 2024), mathematical theorems (Trinh et al., 2024) and/or code syntax & semantics (Ugare et al., 2024). This may also improve efficiency and cost saving. With a stronger LLM, test time compute can be more efficient.”

---

> > > ### Author Response · Authors · 2025-06-24
> > > **Longer version rebuttal to Reviewer Jppn (part 2)**
> > >
> > > (Sorry for any inconvenience!)
> > >
> > > Weakness 3. d)  "We expect RL, evolutionary and neuro-symbolic methods, etc. to play important roles in building agents."
> > >
> > > We want to express that RL and other AI methods are important to build agents. We agree that each AI method may play a role in building agents.  We hope to limit the submission to 12 pages so we do not discuss all AI models in detail.
> > >
> > > We single out the neuro-symbolic method, since it may overcome the limitations of LLMs, or neural methods, e.g., in solving arithmetic problems. We single out the evolutionary method, since it is iterative by nature and it is both a competitor and a collaborator for RL.
> > >
> > >
> > > Weakness 4. a) current LLM agents are agnostic to domain knowledge
> > >
> > > We will make it clearer in the revision that when we say “current LLM agents”, we mean “current popular LLM agents”, e.g., those based on GPT-4, Claude and Gemini, with prompt engineering and manually designed workflows, as well as with fine-tuning or post training. And our point is that when the (popular) LLMs were pre-trained, i.e., following the approach of GPT with next token prediction, they did not leverage domain knowledge.
> > >
> > > Also, although As Reviewer Jppn mentioned that “current state-of-the-art LLMs, such as o3, already stores a good amount of domain knowledge with its pre-training”, the stored knowledge may not be perfect, so that there are errors or hallucinations. And our proposal is to mitigate or even eliminate such errors by incorporating perfect domain knowledge w.r.t. data collection, architecture design and algorithm design, even in the pre-training stage. See also the response to Weakness 3. c) above.
> > >
> > > Weakness 4. b) current LLM agents has large scale of base LLM
> > >
> > > Again, when we say “current LLM agents”, we mean “current popular LLM agents”. See the response to Weakness 4. a)  for more detail.
> > >
> > > When there are smaller, successful models, they are actually the evidence to support our proposal.
> > >
> > > Thanks for pointing to the following papers:
> > > PAE https://arxiv.org/abs/2412.13194
> > > VTool-R1 https://arxiv.org/pdf/2505.19255
> > >
> > > However, both of them are about fine-tuning with RL. We propose to pre-train with iterative improvements. Moreover, their feedback is based on LLMs, and thus not ground truth, as we discuss in Section 5 RQ2: How to attain ground truth data?, in particular, in the 6th paragraph, starting with “Most LLMs are trained with Internet data” and in the 7th paragraph.
> > >
> > > A concrete statement from VTool-R1: “To address this, we use a lightweight LLM-based judge to assess the match between the predicted answer and the ground truth. While not strictly rule-based, this serves as a pseudo rule- based reward appropriate for open-ended tasks such as ChartQA. We reward score of 1 when the judge thinks it is a match.”
> > >
> > >
> > > Weakness 4. c) iterative improvements with RL
> > >
> > > Most LLM agents are based on fixed, imperfect LLMs, with infrequent iterative improvements.
> > > The current hot direction is to use RL for post-training. Some early papers (with RL for fine-tuning):
> > > OFFLINE RL FOR NATURAL LANGUAGE GENERATION WITH IMPLICIT LANGUAGE Q LEARNING https://openreview.net/pdf?id=aBH_DydEvoH
> > > Grounding Large Language Models in Interactive Environments with Online Reinforcement Learning https://proceedings.mlr.press/v202/carta23a/carta23a.pdf
> > > Pangu-agent: A fine-tunable generalist agent with structured reasoning https://arxiv.org/pdf/2312.14878
> > >
> > > If LLM agents are pre-trained with RL, these are evidence supporting our proposal.
> > > There are rare such papers. However, pre-training with RL is promising for future LLM agents, and that is the motivation and purpose of our proposal.
> > >
> > > After our submission, we saw this arxiv paper (June 9, 2025)
> > > Reinforcement Pre-Training, https://arxiv.org/abs/2506.08007
> > > However, it works with GPT next token prediction, for a general model, and does not leverage domain knowledge (enough). There still appears to be a big room, if our proposal is leading to a more efficient and effective direction.
> > >
> > > A couple of earlier papers.
> > > CodeRL pre-trained LLM for code with RL. CodeRL: Mastering Code Generation through Pretrained Models and Deep Reinforcement Learning https://openreview.net/pdf?id=WaGvb7OzySA
> > > In this paper for the game of Diplomacy, RL is part of a loop in pre-training. Human- level play in the game of Diplomacy by combining language models with strategic reasoning
> > > https://www.science.org/doi/10.1126/science.ade9097
> > >
> > > Requested Changes 1 Address the problems in weaknesses: See above.
> > > Requested Changes 2: We will fix inconsistent formats in Section 6.
> > > Requested Changes 3: We will fix the format of Algorithm 1.

---

> > > > ### Comment · Reviewer_Jppn · 2025-06-26
> > > >
> > > > Thanks for your clarification. Here is my response:
> > > >
> > > > 1. (weakness 1. a) and critical point) If AgentZero/$\mu$/$\infty$ are novel frameworks that heavily involves pretraining, then why in Sec. 7.1 you mention that "human player's experience ... is a mixture of Agent$\mu$ and Agent$\infty$"? How can experience itself becomes your framework? Similarly, why a "math or coding agent" is necessarily a mixture of your framework in Sec. 7.3? My guess is that the authors try to say "when you build a game agent / code agent with your framework, it should fall into the category of Agent$\mu$/$\infty$ when doing the 3-way classification of Zero/$\mu$/$\infty$." That being said, the whole Sec. 7 still only consists of some hints (which is not always present - no discussion on how existing works on maths agents can be improved to fit in the proposed framework, and no paper is discussed in Sec. 7.5), instead of definitive and clear insights of suggestions on how to instantiate the proposed framework.
> > > >
> > > > 2. (weakness 1. b)) Please add the refined version of RQ 1.1. The current version in the comment above still misses some part in its logic; for example, why alternative architecture leverages domain knowledge if the underlying training set is all the same?
> > > >
> > > > 3. (weakness 1. c)) The authors mention in their rebuttal that "modular, specialized, likely smaller models ... is feasible and more efficient to iteratively improve models and to achieve better and better models, more frequently than months, likely in weeks or days." **However, the bottleneck here is not the model or algorithm, but the data: How do you get sufficient amount of data that can significantly affect pretraining in less than a few days?** Qwen models use >5 trillion tokens for pretraining (https://arxiv.org/html/2409.12186v3). If these can be significantly updated with high-quality data within a day, how much human effort in curating data will it take - and can Internet provide you with this amount of data at scale within one day?
> > > >
> > > > 4. (weakness 3. a)) Regarding modularity: MoE is an architecture technique and should not fit into the definition of the modularity given in the comment above (treat modularity as specialist models); the "monolithic" models such as Gemini 2.5 pro (https://storage.googleapis.com/deepmind-media/gemini/gemini_v2_5_report.pdf) also uses MoE. Also, I would like to mention that generalizability and complex reasoning (e.g. operate with an entire large code repository) has always been a very important property towards AGI - and small models lack such ability to generalize. A fixed agent workflow of many such small models limits the flexibility and upper bound of the solution.
> > > >
> > > > 5.  (weakness 3. b)) For GPT as a supervised learning method: regardless of whether self-supervised learning can be seen as representation learning or not, the purpose of the whole GPT is to answer the problem directly instead of outputting an embedding. If we view GPT as a representation learning method here, what representation it is trying to learn - and what is the downstream algorithm that uses this representation?
> > > >
> > > > 6. (weakness 3. c)) The discussion only mentions possible solution in a few word without any rationale, instantiation or experiment results. It is hard to believe that the claim is valid without supporting evidence, which I feel is the reason why reviewer GeGt says "the answers for the research questions are too strong and even dubious".
> > > >
> > > > 7. (weakness 2 and 3. d)) The page limit is not a reason for omitting important aspects in the survey; without adequate discussion, claims in this paper may become not well-supported by rationale and proofs and undermine the extent of contribution (which is also pointed out by reviewer GeGt). An example of well-rounded discussion that gets TMLR outstanding certification is https://arxiv.org/abs/2211.09110.
> > > >
> > > > 8. (weakness 4. a)) there is no claim to support that "LLMs did not leverage domain knowledge" since the authors have no access to the underlying dataset of GPT/Gemini. Also, if there is no domain knowledge, how can those LLMs do well without tool use on knowledge-based benchmarks such as GPQA? I suggest the authors to do an experiment to verify whether GPT/Gemini know domain-specific knowledge by testing their performance on domain-specific questions with tool and web search banned.
> > > >
> > > > 9. (weakness 4. b) and 4. c)) One concrete problem of pretraining LLM with RL is that when the model is completely untrained and output random tokens, it is extremely hard to get any reward since there are 150K choice (vocabulary size) for every token and the possible rollout grows exponentially. The paper "Reinforcement Pre-Training" is not really pretraining from scratch; it uses R1-distilled Qwen as the base model and is at best some kind of "post-pretraining". This is also true for the CodeRL paper ("we introduce “CodeRL”, a new framework to improve **pretrained LMs**"), and I am not sure how the diplomacy paper is related.

---

> > > > > ### Comment · Action_Editor_43NQ · 2025-07-03
> > > > >
> > > > > Hi folks,
> > > > >
> > > > > Thanks for your engagement so far. This exchange has gotten pretty deep in the weeds, focusing at times on individual sentences in the paper. Reviewer Jppn, it would be helpful to me (and perhaps to the authors) to have a clearer distinction between significant concerns with respect to the acceptance guidelines and minor, easily remedied concerns about wording etc.
> > > > >
> > > > > The acceptance guidelines focus largely on sufficient support for claims. Given that, can I propose a format that might help focus the discussion?
> > > > > Reviewer Jppn: can you please list what you see as the main claims of the paper? If some key claims are insufficiently supported, can you please indicate what you believe it would take to support that claim?
> > > > > Authors: can you please list what you see as the main claims of the paper? For each one, what support does the paper offer for that claim?
> > > > >
> > > > > Any discrepancies in these two lists will be valuable focus points for clarification. Thanks in advance!

---

> > > > > > ### Author Response · Authors · 2025-07-03
> > > > > > **Do we have to do experiments?**
> > > > > >
> > > > > > Dear Action Editor,
> > > > > >
> > > > > > Thanks for your comments and guidelines.
> > > > > > See the main claims as a reply to "General response to all" on the top.
> > > > > >
> > > > > > We have the impression that Reviewer Jppn is expecting a regular paper, rather than a perspective paper.
> > > > > >
> > > > > > Reviewer Jppn asked about/mentioned experiments twice in the original review and twice in the response to our rebuttal. (See below.)
> > > > > >
> > > > > > We make it clear in the submission that our submission is a perspective paper, and we highlight it again in the Official Comment on top.
> > > > > >
> > > > > > We hope to confirm that, for our submission as a perspective paper, do we have to do experiments?
> > > > > >
> > > > > > We appreciate your time for consideration and clarification.
> > > > > >
> > > > > > Best regards!
> > > > > > The authors
> > > > > >
> > > > > > In Review of Paper4897 by Reviewer Jppn on June 22:
> > > > > >
> > > > > > Weakness 2:
> > > > > >
> > > > > > The contribution of this paper is unclear. The authors made no attempt in building an instance with concrete choice of algorithm components on a particular question is provided in the paper despite discussing research questions in detail, nor do they run any experiment to prove the validity of their framework. Thus, the current claim for adopting the proposed framework is not convincing enough.
> > > > > >
> > > > > > Broader Impact Concerns:
> > > > > >
> > > > > > The paper does not include a broader impact statement. As this paper does not propose any concrete instance of new methods or conduct any experiment, I think the work has no particular ethical concern beyond general LLM works.
> > > > > >
> > > > > > In Official Comment by Reviewer Jppn on June 26:
> > > > > >
> > > > > > Point 6 (weakness 3. c)) The discussion only mentions possible solution in a few word without any rationale, instantiation or experiment results. It is hard to believe that the claim is valid without supporting evidence, which I feel is the reason why reviewer GeGt says "the answers for the research questions are too strong and even dubious".
> > > > > >
> > > > > > Point 8 (weakness 4. a)) there is no claim to support that "LLMs did not leverage domain knowledge" since the authors have no access to the underlying dataset of GPT/Gemini. Also, if there is no domain knowledge, how can those LLMs do well without tool use on knowledge-based benchmarks such as GPQA? I suggest the authors to do an experiment to verify whether GPT/Gemini know domain-specific knowledge by testing their performance on domain-specific questions with tool and web search banned.

---

> > > > > > > ### Comment · Reviewer_Jppn · 2025-07-03
> > > > > > >
> > > > > > > Thank you for your engagement in discussion.
> > > > > > >
> > > > > > > 1. Experiment is one powerful way of proving that the proposed framework can be implemented in applications and acquire benefit. It will be a strong proof if you have the results, but not necessary. My request for the experiment is mainly due to the insufficient discussion and reasoning as I mentioned in other points of weaknesses and responses.
> > > > > > >
> > > > > > > 2. I do not mean to criticize the work for lack of experiment in the "broader impact concerns" section; it's just for evaluation of ethical concern. For weakness 2 and point 6, I put the experiment option besides rationale and instantiation, which means a strong rationale also counts for evidence. For point 8, my suggestion is not to put any experiment result in the paper, but to do a sanity check for the authors themselves on their claim of "whether GPT/Gemini know domain-specific knowledge".

---

> > > > > ### Author Response · Authors · 2025-07-03
> > > > > **Response to Reviewer Jppn (Part 1)**
> > > > >
> > > > > Thanks for the Official Comment.
> > > > >
> > > > > We use the following abbreviations:
> > > > > Jppn0526Comment - Official Comment by Reviewer Jppn on June 26, and
> > > > > Jppn0522Review - Review of Paper4897 by Reviewer Jppn on June 22.
> > > > >
> > > > > Group 1. Some clarifications.
> > > > >
> > > > > Jppn0526Comment, Point 1: “How can experience itself becomes your framework?”
> > > > >
> > > > > It is illustrated in Algorithm 1. It is accessible to people with a background of reinforcement learning. See, e.g., David Silver and Richard Sutton, Welcome to the era of experience, goo.gle/3EiRKIH, 2025, and the RL textbook by Sutton and Barto.
> > > > >
> > > > > Jppn0526Comment, Point 2, “why alternative architecture leverages domain knowledge if the underlying training set is all the same”
> > > > >
> > > > > We gave a clear example in the submission and highlighted it in the rebuttal: The hierarchy of word, statement, function, class, and project in code makes a hierarchical attention mechanism, rather than next token prediction, very promising.
> > > > >
> > > > > Jppn0526Comment, Point 3, “However, the bottleneck here is not the model or algorithm, but the data: How do you get sufficient amount of data that can significantly affect pretraining in less than a few days?”
> > > > >
> > > > > As we discuss in the submission, data, architectures, and algorithms may play important roles in future agents and AI. We propose the general framework, and highlight their importance, and inspire the community to instantiate it and make further progress. Lacking of data may be an issue, but it should be solved, by the community. Check Welcome to the Era of Experience.
> > > > >
> > > > > Jppn0526Comment, Point 5, representation learning
> > > > >
> > > > > Also Jppn25 Weakness 3b) What does it mean by "self-supervised learning to improve the representation, e.g. with GPT as in normal LLMs" (Sec. 3.1)? GPT itself is not a self-supervised learning method and the representation in "normal LLMs" is not learned by a standalone GPT.
> > > > >
> > > > > We treat representation learning as a basic background, which does not necessitate detailed explanations. See e.g., On the Opportunities and Risks of
> > > > > Foundation Models https://arxiv.org/pdf/2108.07258.  Note, in particular in Fig. 2., there is the “adaptation” between “Foundation Model” and “Tasks”.
> > > > >
> > > > >
> > > > > Jppn0526Comment, Point 8, domain knowledge
> > > > >
> > > > > We believe it is a fact: GPT with next token prediction does not leverage domain knowledge.
> > > > >
> > > > > Reviewer Jppn argued something not quite relevant to our submission: that the trained LLMs may exhibit some level of manipulating domain knowledge by achieving decent scores in benchmarks: it is not related to whether a pre-training method leverages domain knowledge or not. Again, consider the example of hierarchy in coding.

---

> > > > > > ### Author Response · Authors · 2025-07-03
> > > > > > **Response to Reviewer Jppn (Part 2)**
> > > > > >
> > > > > > Group 2. We hope a reviewer will be constructive, and avoid assertive statements before thorough discussions.
> > > > > >
> > > > > >
> > > > > > Jppn0526Comment, Point 4, “generalizability and complex reasoning (e.g. operate with an entire large code repository) has always been a very important property towards AGI - and small models lack such ability to generalize. A fixed agent workflow of many such small models limits the flexibility and upper bound of the solution.”
> > > > > >
> > > > > > We treat AGI as an open problem, and there is no conclusive working approach: All LLMs can not guarantee correctness yet. We propose an approach to build strong agents.
> > > > > >
> > > > > > From Jppn, “A fixed agent workflow of many such small models limits the flexibility and upper bound of the solution.”
> > > > > >
> > > > > > We made it clear that our proposal is not about workflow, esp. fixed workflows. RL agents can learn from the environments iteratively.
> > > > > >
> > > > > > Jppn0526Comment, Point 5, “The discussion only mentions possible solution in a few word without any rationale, instantiation or experiment results.”
> > > > > >
> > > > > > We provide three references as below: “We may mitigate or even eliminate hallucinations with domain knowledge during training and/or test stages, e.g., by parsing them out with game rules (Schultz et al., 2024), mathematical theorems (Trinh et al., 2024) and/or code syntax & semantics (Ugare et al., 2024).”
> > > > > >
> > > > > > Jppn0526Comment, Point 7, “The page limit is not a reason for omitting important aspects in the survey; without adequate discussion, claims in this paper may become not well-supported by rationale and proofs and undermine the extent of contribution”.
> > > > > >
> > > > > > From https://jmlr.org/tmlr/, “TMLR caters to the shorter format manuscripts that are usually submitted to conferences, providing fast turnarounds and double blind reviewing.”
> > > > > >
> > > > > > Our submission is a perspective, not a survey, and we believe we do not need to provide detailed explanations about basic background like experience and representation learning.
> > > > > >
> > > > > >
> > > > > > Jppn0526Comment, Point 9, RL
> > > > > >
> > > > > > “One concrete problem of pretraining LLM with RL is that when the model is completely untrained and output random tokens, it is extremely hard to get any reward since there are 150K choice (vocabulary size) for every token and the possible rollout grows exponentially.”
> > > > > >
> > > > > > Reviewer Jppn restricts the discussion to the current popular approach of GPT with next token prediction. Our proposal may not use either Transformer or next token prediction, e.g., consider the example of a potential hierarchical attention mechanism for the hierarchy in coding again. We propose to build specialized agents, so we may not have to encounter the issue of large vocabulary size for NLP problems.
> > > > > >
> > > > > > We hope to express our perspective so that there may be more resources allocated to approaches alternative to popular ones, in particular, (prompt engineering LLMs trained with) GPT with next token prediction.
> > > > > >
> > > > > >
> > > > > > Jppn0522Review, Weakness 2, “The contribution of this paper is unclear. The authors made no attempt in building an instance with concrete choice of algorithm components on a particular question is provided in the paper despite discussing research questions in detail, nor do they run any experiment to prove the validity of their framework. Thus, the current claim for adopting the proposed framework is not convincing enough.”
> > > > > >
> > > > > > Reviewer Jppn’s statement above is assertive yet arguable.
> > > > > >
> > > > > > Jppn0522Review Weakness 3c), “AgentZero/Agent$\mu$/Agent$\infty$ "aims to eliminate hallucinations" (Tab. 1), but the paper does not discuss anything about how this can be achieved (except only briefly mentioned once in RQ2 that knowledge graphs and RAG are imperfect solutions).”
> > > > > >
> > > > > > “but the paper does not discuss anything” is such a strong statement. We provide three reference papers.
> > > > > >
> > > > > > Jppn0522Review Weakness 4 “Some statement in this paper is inaccurate or wrong.”
> > > > > >
> > > > > > 4 a) “This is inaccurate”
> > > > > > 4 b) “this is wrong”
> > > > > >
> > > > > > The papers PAE and VTool-R1 Reviewer Jppn pointed to are not very relevant to our submission, since they are about post-training while we propose to pre-train agents with RL.

---

> > > > > > > ### Comment · Reviewer_Jppn · 2025-07-06
> > > > > > >
> > > > > > > Thank you for your detailed response. Here is my response:
> > > > > > >
> > > > > > > **Group 1**
> > > > > > >
> > > > > > > **Point 1.** According to the authors' explanation, I think this is probably a verbal issue. Both Alg. 1 and the referenced paper discuss about utilizing human experience, but experience itself (i.e. trajectories in RL) is not a framework.
> > > > > > >
> > > > > > > **Point 2.** While the authors provide some examples of leveraging knowledge specific to code being useful, I do not see how Mamba and xLSTM in RQ 3.2 leverages domain knowledge. I also do not see how ViT in RQ 3.2 leverages specific domain knowledge except that they dealt with a input modality different than text. The authors do not provide otherwise connection in the paper for these works and "leveraging domain knowledge".
> > > > > > >
> > > > > > > **Point 3.** I think the issue of data is a legitimate, major and immediate concern of the framework proposed. The "era of experience" paper only mentions a perspective of future agents learning with rich, streamed experience, but the authors make it clear in the rebuttal that "we propose to **pre-train** a model with iterative improvements", which is known to be data-intenstive. The argument "it should be solved by community" is unconvincing to me without a more careful discussion.
> > > > > > >
> > > > > > > **Point 5.** The Fig. 2 provided by the author in the paper does not explain why GPT should be viewed as representation learning. Representation learning is "learning representations of the data that make it easier to extract useful information when **building classifiers or other predictors.**" (https://arxiv.org/pdf/1206.5538) "Adaptation" can also mean finetuning with SFT/RL or in-contect learning on the same model. Unless there is another predictor/classifier installed upon the trained LLM, and the trained LLM itself is not responsible for making decisions, I do not feel "representation learning" is an appropriate term.
> > > > > > >
> > > > > > > **Point 8.** If the authors' argument is that "GPT with next token prediction (as an algorithm) does not leverage domain knowledge is a fact", and the idea is similar to "hierarchy in coding", then I find some arguments in the paper to be inaccurate. For example, Tab. 1 claims that "current LLM agents are agnostic to domain knowledge". It may be true that "their architecture is designed to be agnostic to domain knowledge", but is their algorithm, and especially data, also agnostic? The authors cannot prove this as they have no access to GPT/Gemini data/algorithm, and if it is, I don't believe GPT/Gemini can do well in the benchmarks. The fact that GPT/Gemini works well on domain-specific benchmarks already disproves the claim in Tab. 1 that "current LLM agents are agnostic to domain knowledge".
> > > > > > >
> > > > > > > **Group 2.** In my previous discussion, I disagree with some of the prospects for LLM agents proposed by this paper. While they are not necessarily wrong, as a reviewer, it is my duty to provide my objection as they might also be raised by future readers and hinder the value of the paper if leave unaddressed. I believe revision, clarification, and rationale defending the claims that are reflected in submitting a paper modification against my counterarguments is the most constructive way to enhance this paper.
> > > > > > >
> > > > > > > **Point 4.** Indeed, AGI is an open problem, and there is no standard saying that LLMs should guarantee correctness to achieve AGI, since humans also err. The point is that I hope the authors can add discussions on how the problem in the counterargument can be overcomed, such that the paper better defends itself against different routes to AGI.
> > > > > > >
> > > > > > > **Point 7.** First, it is very often for conference submissions to contain appendices for self-containment of the paper; the idea in the main paper can be precise but the discussion of the whole paper should be well-rounded. Second, the problem is not lack of background knowledge; it is about expanding the discussion of each aspect in greater detail and clearer statements (an example of unclear statement is point 8).
> > > > > > >
> > > > > > > **Point 9.** I hope the authors can expand this idea in detail and put it in revision.
> > > > > > >
> > > > > > > **Weakness 2.** Please feel free to argue against any arguable comments; it is the point of author-reviewer discussion. There is no experiment indeed, and my expectation for "instantiation" is that the authors can provide an discussion of a concrete blueprint of whole system in a standalone section instead of listing papers that accomplishes some of the design; for example, expand the discussion in point 9 in detail. I believe this provides a clearer view for readers to understand the work.
> > > > > > >
> > > > > > > **Point 5 / Weakness 3c).** I don't feel discussion of three papers with each papers using 3-4 words on average is a sufficient rationale or introduction of instantiation of the proposed framework. I would not call it a proper discussion. I hope the authors can extend their discussion in a paper revision.
> > > > > > >
> > > > > > > **4 a) 4 b)** The point is not whether they are relevant to the framework; the point is inaccuracy/overclaim in Tab. 1 for current LLM agents.

---

> > > > > > > > ### Author Response · Authors · 2025-07-07
> > > > > > > > **Complaint about Reviewer Jppn (Part 1)**
> > > > > > > >
> > > > > > > > Dear Action Editor and all,
> > > > > > > >
> > > > > > > >
> > > > > > > > Our submission is a perspective paper: it is not a comprehensive survey, and it is not a 101 level study material for AI, LLM, DL, and/or RL either. As a result, we expect a reviewer or a reader to have a decent background in relevant topics.
> > > > > > > >
> > > > > > > > We appreciate the time Reviewer Jppn spent on our submission. We do hope to have constructive and healthy discussions to improve the submission. However, we found it is very hard to communicate with Reviewer Jppn. This caused lots of confusion, even mental distress to the authors.
> > > > > > > >
> > > > > > > > We feel sorry to do so, but we have to raise the following issues:
> > > > > > > >
> > > > > > > > 1. Reviewer Jppn does not have enough background to review our submission.
> > > > > > > >
> > > > > > > > 2. Reviewer Jppn made many assertive yet arguable or inaccurate statements, before having a decent understanding of our submission, even worse, without enough background.
> > > > > > > >
> > > > > > > > 3. We have an impression that Reviewer Jppn has the tendency of “criticism for the sake of criticism”.
> > > > > > > >
> > > > > > > > 4. We have an impression that Reviewer Jppn is not suitable for reviewing a perspective submission.
> > > > > > > >
> > > > > > > > We believe it should be clear from previous discussions.  We provide some evidence below. We are glad to add more if necessary.
> > > > > > > >
> > > > > > > > We request the Action Editor to consider an alternative plan for the next step, e.g.,
> > > > > > > >
> > > > > > > > 1) deciding based on the discussions so far,
> > > > > > > > 2) seeking comments from Reviewer YbeH and Reviewer GeGt, or even
> > > > > > > > 3) inviting another (qualified) reviewer.
> > > > > > > >
> > > > > > > > We also welcome comments from the community.
> > > > > > > >
> > > > > > > > Best regards,
> > > > > > > >
> > > > > > > > Authors

---

> > > > > > > > > ### Author Response · Authors · 2025-07-07
> > > > > > > > > **Complaint about Reviewer Jppn (Part 2)**
> > > > > > > > >
> > > > > > > > > We use the following abbreviation:
> > > > > > > > >
> > > > > > > > > Jppn0703Comment - Official Comment by Reviewer Jppn on July 3
> > > > > > > > >
> > > > > > > > > Group 1:
> > > > > > > > >
> > > > > > > > > Our Response to Reviewer Jppn (Part 1) on July 3 was about that Reviewer Jppn does not have enough background to review our submission.
> > > > > > > > >
> > > > > > > > > More evidence:
> > > > > > > > >
> > > > > > > > >
> > > > > > > > > Jppn0703Comment, Point 1, “Both Alg. 1 and the referenced paper discuss about utilizing human experience”.
> > > > > > > > >
> > > > > > > > >
> > > > > > > > > Neither Alg. 1 nor the reference is about just “human experience”. This is a clear mistake. Judged from this and previous discussions, Reviewer Jppn does not have a decent understanding of Alg 1, our submission, and the references, David Silver and Richard Sutton, Welcome to the era of experience, goo.gle/3EiRKIH, 2025, and the RL textbook by Sutton and Barto.
> > > > > > > > >
> > > > > > > > >
> > > > > > > > > These references are so fundamental to understand our submission. It is hard to communicate with a reviewer or a reader if they lack such a background (esp. if they are assertive with arguable statements). This is the main reason we decide to raise the issues.
> > > > > > > > >
> > > > > > > > >
> > > > > > > > > Group 2:
> > > > > > > > >
> > > > > > > > >
> > > > > > > > > Our Response to Reviewer Jppn (Part 2) on July 3 was about that Reviewer Jppn made many assertive yet arguable or wrong statements, before having a decent understanding of our submission, even worse, as discussed above, without enough background.
> > > > > > > > >
> > > > > > > > >
> > > > > > > > >
> > > > > > > > > Group 3:
> > > > > > > > >
> > > > > > > > >
> > > > > > > > > We have an impression that Reviewer Jppn has the tendency of “criticism for the sake of criticism”.
> > > > > > > > >
> > > > > > > > >
> > > > > > > > > It is not a constructive and healthy discussion: the authors try to correct mistakes made by a reviewer (trying hard to do so in a respectful way), while the reviewer would make more assertive and arguable or even inaccurate statements.
> > > > > > > > >
> > > > > > > > >
> > > > > > > > > For example, to complement the evidence in issue 1 above, in the Official Comment by Reviewer Jppn on June 26, “How can experience itself becomes your framework?”
> > > > > > > > >
> > > > > > > > >
> > > > > > > > > This is destructive. Even worse, this may result from issue 1 and 2, namely, lack of background and tendency for making assertive yet arguable or wrong statements.
> > > > > > > > >
> > > > > > > > >
> > > > > > > > > We list several pieces of evidence, which may also be evidence for issue 1 and 2 above.
> > > > > > > > >
> > > > > > > > >
> > > > > > > > > 1.
> > > > > > > > >
> > > > > > > > > Jppn0703Comment Group 1 Point 2. “I also do not see how ViT in RQ 3.2 leverages specific domain knowledge”.
> > > > > > > > >
> > > > > > > > >
> > > > > > > > > We treat it as a fact: The design of Visual Transformers (ViT), instead of vanilla Transformers, leveraged the domain knowledge of computer vision.
> > > > > > > > >
> > > > > > > > >
> > > > > > > > > 2.
> > > > > > > > >
> > > > > > > > > In Official Comment by Reviewer Jppn on June 26, “generalizability and complex reasoning (e.g. operate with an entire large code repository) has always been a very important property towards AGI - and small models lack such ability to generalize.”
> > > > > > > > >
> > > > > > > > >
> > > > > > > > > This is an assertive and arguable or inaccurate statement.
> > > > > > > > >
> > > > > > > > >
> > > > > > > > > In Jppn0703Comment Group 1 Point 4, Reviewer Jppn agreed with our rebuttal “AGI is an open problem”, and then made more arguments and more requests.
> > > > > > > > >
> > > > > > > > >
> > > > > > > > > 3.
> > > > > > > > >
> > > > > > > > > Jppn0703Comment Group 1 Point 5. About representation learning.
> > > > > > > > >
> > > > > > > > > We believe when we mention “adaptation”, it should be sufficient for a reader or a reviewer with a decent background in deep learning to understand “representation learning”.
> > > > > > > > >
> > > > > > > > >
> > > > > > > > > 4.
> > > > > > > > >
> > > > > > > > >
> > > > > > > > > Jppn0703Comment, last paragraph, “the point is inaccuracy/overclaim in Tab. 1 for current LLM agents”.
> > > > > > > > >
> > > > > > > > >
> > > > > > > > > GPT with next token prediction is agnostic to domain knowledge, although it is true that data do contain domain knowledge. We provided an example, “The hierarchy of word, statement, function, class, and project in code makes a hierarchical attention mechanism, rather than next token prediction, very promising.” It turned out that Reviewer Jppn responded with “inaccuracy/overclaim”.
> > > > > > > > >
> > > > > > > > >
> > > > > > > > >
> > > > > > > > > Group 4:
> > > > > > > > >
> > > > > > > > >
> > > > > > > > > We have an impression that Reviewer Jppn is not suitable for reviewing a perspective submission.
> > > > > > > > >
> > > > > > > > >
> > > > > > > > > Reviewer Jppn asked for more detail than a perspective paper should provide, sometimes based on inaccurate understanding of the submission and/or lack of background knowledge.
> > > > > > > > >
> > > > > > > > >
> > > > > > > > > In the Official Comment by Reviewer Jppn on July 3, “Experiment … not necessary.”
> > > > > > > > > If experiments are not necessary, it may not be proper for a reviewer to request authors for them, since this increases the confusion. Instead, ask authors to improve writing or just reject the submission.
> > > > > > > > >
> > > > > > > > > Our title is: Iterative Improvements Based on Ground Truth: Building LLM Agents in the Era of Experience Inspired by Games AI. Reviewers and readers should know that it is likely infeasible for resource-poor researchers to implement such an idea. Moreover, it is a perspective for a general framework: a specific implementation for one problem, e.g., maths agent, may not be sufficient to validate the general framework for all problems, esp. according to Reviewer Jppn’s logic.

---

> > > > > > > > > > ### Comment · Reviewer_Jppn · 2025-07-07
> > > > > > > > > >
> > > > > > > > > > Thank you for your follow-up, and thanks for AE for the time. It is regrettable to see that the authors are:
> > > > > > > > > >
> > > > > > > > > > 1. Not willing to provide revision to their paper after discussion (e.g. "it is likely infeasible for resource-poor researchers to implement such an idea" -> how can writing a detailed blueprint consuming too much resource?; the authors do not take time to provide updated RQ 1.1; arguing "then made more arguments and more requests" -> addressing the reviewer's concern is part of the authors' duty).
> > > > > > > > > >
> > > > > > > > > > 2. Refuse to modify the misleading words in their paper ("current LLM agents are not domain-aware"), and inconsiderate of the readers (refuse to add explanations to make the paper self-contained, or adding how the interpretation of their work should be in their paper), and not directly respond to my questions (e.g. the issue of data and inaccurate words in Tab. 1). Of all the TMLR papers I have reviewed, the authors of this paper is the least willing to update their paper despite the long discussion time and question from other reviewers GeGt.
> > > > > > > > > >
> > > > > > > > > > As the authors are unwilling to continue the discussion, I will not further update my opinion on this paper and will vote for rejection. I believe there is a misunderstanding in some aspects (e.g. the definition of "domain knowledge"), but it is unlikely to reach a consensus with the authors given the current situation. I will leave the decision to AE for whether a new reviewer should be invited and whether this review should be ignored.

---

> > > > > > > > > > > ### Comment · Reviewer_Jppn · 2025-07-07
> > > > > > > > > > >
> > > > > > > > > > > p.s. "... and question from other reviewers GeGt." -> "and provide revision submission based on question from other reviewers such as GeGt."

---

### Review · Reviewer_YbeH · 2025-06-22

**Summary Of Contributions:**

This paper is a perspective paper and proposes a roadmap for building LLM agents from two main aspects: 1) iterative improvement of agents by feedback and learning, and 2) the source of ground-truth data from oracle model, verifier or real-world interaction. The authors ask three research questions on domain knowledge usage in agent, the source of ground truth data, and iterative improvements. The authors argue that the current LLM agents lack true autonomy and optimality. Meanwhile, they advocate for domain-specific, small, iteratively-trained agents guided by strong ground truth data sources.

**Audience:**

Yes

**Claims And Evidence:**

Yes

**Requested Changes:**

I don't have any request on paper revision. This paper is to the interest of LLM community and TMLR audience.

**Strengths And Weaknesses:**

**strengths**
- The paper is clearly written and easy to follow. It uses the example of literature in games AI to illustrate the idea of iterative improving and the fidelity of the world model, making it more intuitive and easier to understand for the audiences.
- The proposed questions on how to develop LLM agents cover two key elements in LLM: where are the data from and how to utilize those data by algorithm and model architecture, which are crucial to the construction of future model (e.g., superhuman intelligence) and of the interest of many researchers in the community.

**weaknesses**
There is no major weakness in this paper. As this paper is a perspective paper, the main objective is to propose a roadmap and this is no grouth truth answer on whether the statement is correct or not.

---

> ### Author Response · Authors · 2025-06-24
> **We appreciate the supportive review by Reviewer YbeH.**
>
> We will refine the draft, esp. as suggested by the other reviews, and according to new progress in the field, e.g., AlphaEvolve: A Gemini-powered coding agent for designing advanced algorithms, https://deepmind.google/discover/blog/alphaevolve-a-gemini-powered-coding-agent-for-designing-advanced-algorithms/.

---

### Author Response · Authors · 2025-06-24
**General response to all**

Part 1 about contributions
Part 2 for a brief summary for RQ1.1: How to leverage domain knowledge?
Part 3 What is modularity for LLMs?

Part 1 about contributions

This is a perspective paper. It is about a blueprint for a potential, promising, near future solution. Our proposal is in stark contrast to the current popular approaches like LLM-based agents with prompt engineering and workflows. Ideally, it is about pre-training, so that there are chances to build better base models, and make fine-tuning and prompting more efficient and effective. Thus our proposal is different from the current popular approaches about post-training. On the other hand, as a more “practical”, interim solution, our proposal may be adapted to work in a post-training style.

A full implementation of our proposal should work at the pre-training stage, considering not only data collection and algorithm design, but also architecture design. That is, we may consider alternatives to (vanilla) Transformers and next token prediction. Different (neural network) architectures may be more suitable to different problems, e.g., as we discuss for RQ3.2: What are limitations of current LLMs?: “Wang et al. (2023) apply discontinuous networks for mathematics (Della Santa & Pieraccini, 2023) to contract design.”

Besides the contributions as shown below, we should also highlight that we propose to pre-train with interactive improvements using RL, e.g., with enough resources, as presented in Algorithm 1, i.e., the whole Algorithm 1 is ideally working in the pre-training stage. We will make this clearer in the revision. (Again, our proposal may work in a post-training style.)

We make the following contributions for building LLM agents: 1) We highlight the importance of both iterative improvements and ground truth. 2) We highlight the importance of domain knowledge, w.r.t. data collection, architecture design and algorithm design, and to strike a balance between a) scale and b) performance, efficiency and cost. 3) We expect to help improve the interpretability, trustworthiness and safety. 4) We briefly survey relevant works, make a critical examination, and propose a potential solution.

Part 2 for a brief summary for RQ1.1: How to leverage domain knowledge?

Moreover, we plan to add a (sub)section for a brief summary to answer RQ1.1: How to leverage domain knowledge?, likely in the beginning of Section 8 Compare with current approaches, and refer to it when we mention RQ1.1 in Section 4. A draft follows and we will definitely refine it in the revised version.
In the above, we discuss RQ1.1: How to leverage domain knowledge? when we discuss data collection, architecture design and algorithm design. Here is a brief summary. In Algorithm 1, collecting ground truth data, designing auxiliary tasks, and even collecting experience, may require domain knowledge. In Section 5 RQ2: How to attain ground truth data?, we also discuss the question: Are synthetic data helpful?. Domain knowledge is essential for a perfect world model (e.g., game rules), a perfect verifier (e.g., maths and coding), or a high-fidelity simulator (e.g., physical laws underlying robotics simulators), thus for data collection.  When we discuss RQ3.2: What are limitations of current LLMs?, we mention alternative neural network architectures, e.g., TreeLSTM, GNN, discontinuous networks for mathematics (Della Santa & Pieraccini, 2023), contract design (Wang et al., 2023), Visual Transformers (ViT) for image recognition (Dosovitskiy et al., 2021), multi-action-value Transformer model for board games (Schultz et al., 2024), and different levels of abstraction, e.g., CodeBPE tokenization for code (Chirkova & Troshin, 2023). All these leverage domain knowledge. The architecture may influence algorithms. In Sections 7.1 Games agents, 7.2 Maths agents, and 7.3 Coding agents, we present concrete examples for how to leverage domain knowledge, in particular, for coding, the precise domain knowledge of Abstract Syntax Tree (AST), control flow graphs (CFG), data flow graphs (DFG), and compiler intermediate representation. The hierarchy of word, statement, function, class, and project in code makes a hierarchical attention mechanism, rather than next token prediction, very promising.

Part 3 What is modularity for LLMs?

“We propose to build specialist models, rather than generalist LLMs. We propose to divide and conquer, rather than following a holistic approach.” in Section 8 Compare with current approaches.
We basically treat modularity as specialist models, in a stark contrast to the current popular approach that LLM agents are based on a monolithic LLM like GPT-4. It is already a big topic and may cause lots of discussions or even debates, so we do not intend to discuss finer-granularity of modularity for a single specific agent, e.g., there would be lots of sub-modules for a coding agent, e.g., specification, generation, compiling, testing, verification, security, safety, UI/UX, etc.

---

> ### Author Response · Authors · 2025-07-03
> **Main claims**
>
> Dear Action Editor, Reviewers, and all Readers,
>
> We use the first two contributions as our main claims.
> Since it is a perspective paper, we mainly use “reasoning” based on our understanding of relevant background knowledge and sometimes educated guesses, and we do not provide empirical results (since we do not have enough resources to implement our proposal and  conduct decent experiments). See also “Part 1 about contributions”  in “General response to all” on the top, where we also highlight pre-training.
>
> We hope to express our perspective so that there may be more resources allocated to approaches alternative to currently popular ones, in particular, (prompt engineering LLMs trained with) GPT with next token prediction.
>
> In the following, we list Research Questions (RQ), and we provide supporting evidence in the corresponding sections. Algorithm 1, Table 1 and Table 2 illustrate our perspective.
> We are happy with answering more questions.
>
> 1) We highlight the importance of both iterative improvements and ground truth.
>
> RQ2: How to attain ground truth data?
>
> RQ3: How to make iterative improvements? RQ3.1: Why are iterative improvements important? RQ3.2: What are limitations of current LLMs? RQ3.3: Will scaling up current LLMs achieve reasoning and planning capacity to build agents? RQ3.4: Large vs small models? RQ3.5: Modularity? Generalist vs specialist? RQ3.6: What algorithms are suitable for iterative improvements?
>
> 2) We highlight the importance of domain knowledge, w.r.t. data collection, architecture design and algorithm design, and to strike a balance between a) scale and b) performance, efficiency and cost.
>
> RQ1: Should we leverage domain knowledge? RQ1.1: How to leverage domain knowledge?
>
> In the paper: We tackle RQ1.1: How to leverage domain knowledge? when we discuss data collection, architecture design and algorithm design.
>
> We provide a brief summary for RQ1.1: How to leverage domain knowledge? in “General response to all” on the top, Part 2.
>
> Best regards!
> Authors

---

> > ### Author Response · Authors · 2025-07-10
> > **Major changes in revision (Part 1)**
> >
> > Dear Action Editor, Reviewers, and all readers,
> >
> > It appears that we do not have to revise the PDF before we know the acceptance decision.
> > However, a couple of days ago, we realized that we could upload a revision and it would be convenient for reviewers and the Action Editor.
> >
> > We have made refinements here and there, adding some references.
> > Below we list major changes.
> >
> > When we have more time and when we collect more feedback, we will make more thorough improvements.
> >
> > Best regards,
> >
> > Authors
> >
> > 1.
> >
> > Added the third last paragraph in Introduction:
> >
> >
> > Our paper is about a blueprint for a potential, promising, near future solution. Our proposal is in stark contrast to the current popular approaches like LLM-based agents with prompt engineering and workflows. Ideally, a full implementation of our proposal should work at the pre-training stage, considering not only data collection and algorithm design, but also architecture design. That is, we may consider alternatives to (vanilla) Transformers and next token prediction. Different (neural network) architectures may be more suitable for different problems, as we discuss for RQ3.2 in Section 6. As a result, there are chances to build better base models, and make fine-tuning and prompting more efficient and effective. Thus our proposal is different from the current popular approaches about post-training. On the other hand, as a more “practical”, interim solution, our proposal may be adapted to work in a post-training style. We hope to express our perspective so that there may be more resources allocated to approaches alternative to currently popular ones, in particular, LLM-based agents following the pipeline: 1) pre-training an LLM using GPT with next token prediction by some resource-rich organization, then, 2) relying on the fixed LLM, using prompts, workflows, etc. to build agents by the community.
> >
> > More explanation for “Otherwise, it would be very long”, which is not in the revision: If we need to explain “representation learning” and “learning from experience”, then there are too many concepts waiting for explanations: imitation learning, RLHF, inverse RL, meta-learning, decision-time planning, auxiliary learning, MCTS, formal verification, model predictive control (MPC), just name a few. Value function, policy, reward function, SSL, etc. may also be on a similar level. Moreover, we do touch many disciplines, like game theory, operations, research, optimal control, etc.
> >
> > Consequently, we as authors believe we should not try to please a reviewer by providing detailed explanations about “representation learning” and “learning from experience”, which are basic background for the submission.
> >
> > 2.
> >
> > In Section 3.1 AgentZero: AlphaZero-like LLM agents, second paragraph, add the reference Bommasani et al., 2022, On the Opportunities and Risks of Foundation Models, and the statement “to facilitate adaptation for further training” for  representation learning.
> > In Stage 1 for representation learning (Bommasani et al., 2022), we employ self-supervised learning (SSL), e.g., GPT as in normal LLMs, to facilitate adaptation for further training, in particular, for RL training.
> >
> > 3.
> >
> > We did not expect so much discussion/confusion about “agnostic to domain knowledge”. We treat it as a basic background or a common sense. However, we added the following in the caption of Table 1 to facilitate much easier understanding.
> > Note, GPT with next token prediction is agnostic to domain knowledge, although data do contain domain knowledge. For example, as we will discuss in Section 7.3, when a code LLM is trained using GPT with next token prediction, it does not leverage perfect code domain knowledge, like abstract syntax tree (AST), data flow graph (DFG), control flow graph (CFG), and the hierarchy of word, statement, function, class, and project in code.
> >
> > We also added:
> >
> >
> > When we say “current LLM agents”, we mean “current popular LLM agents”, in particular, those based on fixed LLMs, with manually designed prompts and workflows.
> >
> >
> > 4
> >
> > In Section 3.2 Agent_\mu: MuZero-like AI agents, second paragraph, add
> >
> >
> > and also earlier work about cooperative inverse RL (Hadfield-Menell et al., 2016) on alignment. We may need to consider reward misspecification and reward hacking, e.g., inverse reward design (Hadfield-Menell et al., 2017).

---

> > > ### Author Response · Authors · 2025-07-10
> > > **Major changes in revision (Part 2)**
> > >
> > > 5.
> > >
> > > In Section 4 RQ1: Should we leverage domain knowledge?
> > >
> > >
> > > Updated first couple of sentences as:
> > >
> > > Precise, rich and valuable domain knowledge implies strong prior information or strong inductive bias, thus leading to more efficient and effective learning systems. It is thus desirable to leverage such information in contrast to an AI agnostic to, or not leveraging enough, domain knowledge.
> > >
> > > Added:
> > >
> > > Sutton (2019) advocates a meta-method to incorporate human knowledge. However, it appears that the discussion in Sutton (2019) does not include the case for perfect domain knowledge. This leaves a question: Should we learn such domain knowledge, e.g., axioms and those derived from them? We argue, similar to Brooks (2019) and Kaelbling (2019), but for perfect domain knowledge, that it is desirable to incorporate it in data collection, architecture design and algorithm design to achieve more efficient and effective learning.
> > >
> > > Added:
> > >
> > > We should differentiate the research questions about "Should we" and "How to" leverage domain knowledge. We should leverage domain knowledge as much as possible  to exploit the strong and perfect prior information. However, how to leverage domain knowledge is problem dependent, e.g., it is relatively easier for games, maths and coding, and it may be non-trivial for problems with human-in-the-loop.
> > >
> > > 6.
> > >
> > > Added fourth paragraph for RQ3.2: What are limitations of current LLMs?
> > >
> > > György et al. (2025) propose exact learning for deductive reasoning, which requires correctness on all in- puts, rather than optimizing statistical performance w.r.t. a distribution. This is in stark contrast to the current statistical learning paradigm for LLMs, and may demand novel designs for data collection, (neural) architecture and algorithm.
> > >
> > > 7.
> > >
> > > Added last paragraph in RQ3.4: Large vs small models? RQ3.5: Modularity? Generalist vs specialist?
> > >
> > > We basically treat modularity as specialist models, in stark contrast to the current popular approach that LLM agents are based on a monolithic LLM. It is already a big topic and may cause lots of discussions or even debates, so we do not intend to discuss finer-granularity of modularity for a single specific agent, e.g., there would be lots of sub-modules for a coding agent, e.g., specification, generation, compiling, testing, verification, security, safety, UI/UX, etc.
> > >
> > > Added a reference:
> > >
> > > Belcak et al. (2025) argue that SLMs are powerful, more suitable, and more economical for agents.
> > >
> > > Small Language Models are the Future of Agentic AI
> > > https://arxiv.org/abs/2506.02153
> > >
> > > 8.
> > >
> > > In RQ3.6: What algorithms are suitable for iterative improvements?
> > >
> > > We expect RL, evolutionary and neuro-symbolic methods, among other AI methods, to play important roles in building agents. Neuro-symbolic methods may overcome the limitations of LLMs, or neural methods, e.g., in solving arithmetic problems. Evolutionary methods are iterative by nature and are both competitive and collaborative with RL, e.g., FunSearch (Romera-Paredes et al., 2024) and AlphaEvolve (Novikov et al., 2025).
> > >
> > >
> > > Additional Comment 1: Collecting data may be an open problem.
> > >
> > > At the end of 5 RQ2: How to attain ground truth data?, we mention: “However, there are still questions to tackle. How to deal with limited ground truth data? How to collect experience efficiently? How to design efficient learning algorithms, esp. with limited experience? We leave them as open problems.”
> > > We propose the approach of iterative improvements based on ground truth to building agents. And we acknowledge collecting data may be an open problem. It is natural for a perspective paper to have many open problems in initialization, so that we are not able to provide details on how to instantiate all, even some of them. As authors of a perspective paper, we have to leave the open problems to the community.
> > >
> > > Additional Comment 2:  Tone
> > >
> > > We are glad to adjust the tone of the paper. However, as authors, we are confident with our submission. We appreciate more details about where we should adjust.

---

> ### Comment · Reviewer_Jppn · 2025-07-03
> **Summary of Concerns**
>
> Dear Action Editor,
>
> Thanks for your effort in making decision for this paper. The concerns that I mention in my review are all major concerns (except requested changes 2 and 3) that prevents me from recommending acceptance of this paper. My second round of response discusses about every weakness that I mention in my review, as the author's response does not address my concern.
>
> Here I briefly summarize my concerns:
>
> 1. The paper discusses many types of application in Sec. 7, such as game agent or code agent. The paper discusses about relevant works, but lack definitive and clear insights of suggestions on how to instantiate the proposed framework except for brief mentioning  sometimes in one or two sentence (e.g. "by using special architecture"), which I feel is an insufficient discussion. Similar concern also holds for the discussion of "elimination of hallucination", and some other claims in this paper. The authors explained that their limited discussion on several topics is due to the 12-page limit, which I feel is not a valid cause; the quality of the paper is the decisive factor.
>
> 2. RQ 1.1, "how to leverage domain knowledge", is not properly discussed. The authors have not submitted their refined version of this section, and the current section still has reasoning concern (see my response to authors).
>
> 3. "Modularity" in this paper is not properly discussed. Some of the examples of modularity (e.g. MoE) in the paper contradicts with the author's explanation in the rebuttal. Also, I feel the modularity solution proposed in the paper is unrealistic for lack of data. I mentioned this concern in my response and the authors have not responded.
>
> 4. The idea of seeing LLM as "self-supervised learning **that learns representations**" in Alg. 1 is not thoroughly explained.
>
> 5. The authors claim that "monolithic LLMs did not leverage domain knowledge" and "use RL for pretraining", which I feel is not convincing and probably incorrect. I give my reason in my response and the authors have not answered. The paper evidence provided by authors in the rebuttal is not valid (see my response).

---

> > ### Author Response · Authors · 2025-07-03
> >
> > Thanks Reviewer Jppn for Summary of Concerns.
> > See response below.

---

> ### Comment · Reviewer_Jppn · 2025-07-12
>
> Thanks the authors for updating the manuscript. I appreciate that
>
> 1. The paper now makes it clear that the proposed framework should work at the pre-training stage with alternative architecture.
> 2. The paper now uses a more objective tone in the answer to RQ1.
> 3. Now the definition of “modularity” in this paper is clearer. I would suggest adding some sentence in RQ3.4 like “modularity has different definitions, e.g. Stoica et al. Here, we basically treat modularity as …” nearer to the beginning RQ 3.4.
>
> Below are my remaining concerns:
>
> 1. Regarding “otherwise, it would be very long”: I never asked the authors to add a preliminary section to explain the notion of representation learning. What I asked, as mentioned in my Jun. 26 response, point 5, is to explain in one sentence in Sec. 3.1, “What will be the input for the downstream task with or without this representation learning, i.e. What is the raw input and what is its representations?” Take RL as an analogy: for an RL paper, it is reasonable to assume the readers know the notion of reward function, but the paper should specify what is the input and what is its correspondence to output of the reward function in the context. And while I read the related paper provided by the authors (in Jul. 6 response), I do not believe that most readers from the general audience would take time to find related information in a provided link to another paper.
>
> 2. Regarding Tab. 1, it is good that the authors specify “although data do contain domain knowledge” and give examples in Tab. 1. After a long discussion, I understand that the authors try to convey the message “current, mainstream LLM agents are designed for general tasks, and not considering domain knowledge such as code structure in its design”. However, there are tens of thousands of LLM paper coming out each year (https://arxiv.org/abs/2504.09037), and most readers do not have the patience; instead, they only have time to skim through Tab. 1, “current LLM agents are agnostic do domain knowledge”, find it arguable as I have mentioned, and may ignore the rest of the paper. Therefore, the authors should be extra careful on the summary table. A minimum fix is to change “agnostic to domain knowledge” in Tab. 1 to “agnostic to domain knowledge in their design”.
>
> 3. Regarding RQ1: The authors added a sentence “Sutton (2019) advocates a meta-method to incorporate human knowledge”. Could the authors also add a sentence to briefly explain what it is? Also, could the authors add the section link on “when we discuss data collection, architecture design and algorithm design”? Also, if the summary is a single sentence “In Algorithm 1, collecting ground truth data, designing auxiliary tasks, and even collecting experience, may require domain knowledge”, I think there is no need to add “here is a brief summary” in Sec. 8. Slightly expanding the summary in Sec. 8 would be better.
>
> 4. Now that the paper made it clear that small models (and / or partial update of large models) is more desirable and use the word “SLM” for $\leq 7B$ language model. However, the title, abstract and introduction are all around “LLM” agents. It is true that “SLM”s are also treated as “LLM”s generally; however, since the authors also propose to use different architecture and treat models as specialists, the lower bound for the definition of “LLM” in this paper becomes a bit unclear. A small LSTM-based sentiment classifier with <1M parameters is also a specialist, a language model, and can be iteratively updated. But is it an LLM? What is the “minimum requirement” for models that can be called as LLM in this paper? A fix might be to add a brief discussion on the granularity of modularity / specialist.
>
> 5. In RQ 3.6, the authors add “Evolutionary methods are iterative by nature and are both competitive and collaborative with RL” and provide some references. It would be better if “how they are competitive / collaborative” in the paper can be summarized in this paper briefly for these papers.
>
> 6. I still feel the authors should try to propose something, albeit possibly not in detail or without implementation, to address the pre-training data issue. It’s fine if the authors insist this is too much for a perspective paper; I will leave the decision for whether this affects the strength of the message of the work to AE.
>
> A few advice:
> 1. Next time when providing revision, it is good to mark the revised part in the pdf in a different color such that it is easier to locate the modifications.
> 2. It is good to be confident with submission from authors themselves (and they should). However, the confidence may silently lead the authors into the “curse of knowledge” that make them overlook possibly misleading or arguable statements. Reviewers are already careful readers, and thus authors should be even more careful as the paper will be eventually tested by the general audience as readers.

---

> > ### Author Response · Authors · 2025-07-14
> > **Response to Reviewer Jppn (Part 1)**
> >
> > Thanks for the feedback by Reviewer Jppn
> >
> > Comment 3. Modularity
> >
> > Plan to update:
> >
> > “Stoica et al. (2024) discuss that modularity is critical for building reliable systems, by decomposing a system into components and combining them together, with examples …”
> >
> > “As discussed above, modularity may have different definitions, like those in Stoica et al. (2024) and MoE. Following Stoica et al. (2024) and different from MoE, we basically treat modularity as specialist models, …”.
> >
> > FYI, an excerpt from Stoica et al. (2024), “Modularity: The ability to decompose a system into multiple components and combine them together.”
> >
> > Concern 1. Representation learning
> >
> > In Bommasani et al., 2022, in Introduction for Deep Learning, there is “Deep neural networks would be trained on the raw inputs (e.g., pixels), and higher-level features would emerge through training (a process dubbed “representation learning”).”
> >
> > The first sentence in Representation Learning: A Review and New Perspectives, 2014, https://arxiv.org/pdf/1206.5538 is: “The performance of machine learning methods is heavily dependent on the choice of data representation (or features) on which they are applied.”
> >
> > The first sentence in the wiki entry for “”Feature learning” is: “In machine learning (ML), feature learning or representation learning is a set of techniques that allow a system to automatically discover the representations needed for feature detection or classification from raw data.” (when googling “representation learning”)
> >
> > The output of a representation learning depends on the input, i.e., different input with text, image, vidoe, code, or other modalities. and their combinations, may generate different representations/features/outputs.
> >
> > We plan to update “In Stage 1 for representation learning (Bommasani et al., 2022)” to “In Stage 1 for representation learning, i.e., feature learning (Bengio et al. 2013, Bommasani et al., 2022),”
> >
> > Bengio, Yoshua, Aaron Courville, and Pascal Vincent. "Representation learning: A review and new perspectives." IEEE transactions on pattern analysis and machine intelligence 35.8 (2013): 1798-1828.
> >
> > Concern 2. Domain knowledge
> >
> > We will change it to “GPT with next token prediction by design is agnostic to domain knowledge”
> >
> > Concern 3. Meta-methods
> >
> > Plan to update:
> >
> > “Sutton (2019) advocates meta-methods to incorporate human knowledge, by learning them, rather than building them in by humans.”
> >
> > FYI, the last paragraph in The Bitter Lesson about meta-methods, http://www.incompleteideas.net/IncIdeas/BitterLesson.html
> > Note, we discuss the difference between human knowledge and (perfect) domain knowledge.
> >
> > Concern 4. Small language model
> >
> > Plan to add something like:
> >
> > We propose to build specialized, modular, small language models for agents. Aligning with the current trend of LLMs, we choose to build agents with language model. We hope the language models to be as small as possible, on the condition of achieving a certain level of competence.
> >
> > The last sentence in the third paragraph in Section 8 Compare with current approaches: “We propose to build many specialized agents, with one or many coordinator agents, and then form the multi-agent system with specialists, making iterative improvements based on ground truth, in contrast to the current popular approach in which agents are based on general, fixed, imperfect LLMs.”
> >
> > Concern 5. Evolutionary methods
> >
> > Plan to update:
> >
> > Evolutionary methods are iterative by nature and are both competitive, e.g., FunSearch (Romera-Paredes et al., 2024) and AlphaEvolve (Novikov et al., 2025), and collaborative, e.g., Zheng et al (2019),  with RL.
> >
> > Zheng, Yan, et al. "Wuji: Automatic online combat game testing using evolutionary deep reinforcement learning." 2019 34th IEEE/ACM International Conference on Automated Software Engineering (ASE). IEEE, 2019.

---

> > > ### Author Response · Authors · 2025-07-14
> > > **Response to Reviewer Jppn (Part 2)**
> > >
> > > Concern 6. data collection
> > >
> > > In the current submission, we have the following about data collection.
> > >
> > > Section 5 RQ2: How to attain ground truth data?
> > >
> > > We tackle RQ1.1: How to leverage domain knowledge? when we discuss data collection, architecture design and algorithm design, and present a brief summary in Section 8.
> > >
> > > It is true we do not provide a concrete pseudo code or a detailed instantiation plan for data collection. However, we do provide some detailed explanation with many references, e.g., in Section 7.3 Coding agents. We have the following two statements:
> > >
> > > “A code model may operate at multiple levels of abstraction, from word, statement, function, class, to project, which calls for more studies.”
> > >
> > > “There are papers exploiting code domain knowledge for LLMs, like Abstract Syntax Tree (AST), control flow graphs (CFG), data flow graphs (DFG), and compiler intermediate representation, with trees and/or graphs, …”.
> > >
> > > Note, data collection is problem dependent, and it is in general an open problem (esp for a perspective paper), as we admitted in the submission: “However, there are still questions to tackle. How to deal with limited ground truth data? How to collect experience efficiently? How to design efficient learning algorithms, esp. with limited experience? We leave them as open problems.”
> > >
> > > We are glad to provide more details, e.g., a brief plan for data collection for  coding or maths based on SOTA and our educated guess, esp. if the Action Editor and reviewers regard this as necessary, and this is the right way to respond.
> > >
> > > We do hope to know how much more details and in which ways we should explain for data collection, in particular, for coding agents, beyond those in Section 7.3 Coding agents, and those summarized again in Section 8 Compare with current approaches.
> > >
> > > “In Sections 7.1 for games agents, 7.2 for maths agents, and 7.3 for coding agents, we present concrete examples for how to leverage domain knowledge, in particular, for coding, the precise domain knowledge of Abstract Syntax Tree (AST), control flow graphs (CFG), data flow graphs (DFG), and compiler intermediate representation. The hierarchy of word, statement, function, class, and project in code makes a hierarchical attention mechanism, rather than next token prediction, very promising.”
> > >
> > > We provide concrete suggestions on what types of data to collect for building coding agents: AST, CFG, DFG, compiler intermediate representation, and the hierarchy of code, as well as a potential, concrete architecture design: hierarchical attention.
> > >
> > > Such a detailed, illustrative explanation, together with Algorithm 1 (and more explanations in the submission), provides a rather clear research idea; and for resource-rich people, it may be worth trying to see how it compares with current coding LLMs. Admittedly, there are many papers to refer to when designing  data collection, architectures and algorithms.
> > >
> > > Advice 2
> > >
> > > In our last response: We are glad to adjust the tone of the paper. We appreciate more details about where we should adjust.

---

> > > > ### Comment · Reviewer_Jppn · 2025-07-19
> > > >
> > > > Thanks for your response. Here are my follow-ups:
> > > >
> > > > 1. As the authors mentioned, the input is the "raw" input, which could be text, image or audio (depending on the modality of LLM). However, I think the definition for "higher level feature" here is still unclear. Is it a structured text, a feature map or something similar? The answer to this question might vary as the authors consider alternative architectures, but it is still worthwhile clarifying.
> > > >
> > > > 2 to 5. I appreciate the authors' modification, and looking forward to see a revised version of the paper.
> > > >
> > > > 6. I appreciate the author's discussion. Can these be summarized somewhere in the paper?
> > > >
> > > > 7. I will leave the question of "where to adjust specifically" to reviewer GeGt.

---

> > > > > ### Author Response · Authors · 2025-07-20
> > > > > **Thanks Reviewer Jppn**
> > > > >
> > > > > We plan to add the following, likely as a footnote:
> > > > >
> > > > > “The output of a representation learning depends on the input, i.e., different input with text, image, video, code, or other modalities. and their combinations, may generate different representations, features, or outputs, i.e., embeddings of the input.”
> > > > >
> > > > >
> > > > > We guess that adding “feature learning” as an explanation to “representation learning” should be sufficient for a TMLR paper, in particular, a perspective paper on RL and LLMs agent.
> > > > >
> > > > > Otherwise, further clarification may take a long paragraph, discussing potential input output pairs, which would become distractive to readers.
> > > > >
> > > > > However, we are glad to figure out what may be a proper way to clarify “representation learning” further, if Action Editor and reviewers think it is critical.

---

> > > > > > ### Comment · Reviewer_Jppn · 2025-07-21
> > > > > >
> > > > > > I think one example in this "representation learning output" with corresponding is suffice (e.g. in xxx task, input is image and output is convolutional map); there is no need to add a long paragraph, as the point is to give readers a general impression on how this works.

---

> > > > > > > ### Author Response · Authors · 2025-07-21
> > > > > > > **Thanks Reviewer Jppn**
> > > > > > >
> > > > > > > We will revise the submission as suggested by you.

---

### Decision · Action_Editor_43NQ · 2025-08-13

**Recommendation:** Reject

**Additional Comments:**

Because of the challenging author-reviewer interaction, and partly at the authors' request, I have incorporated more of my own evaluation of the paper into my final decision than I normally would. I have done my best not to replace the reviewers' critique and recommendations but to use my own reading of the paper to contextualize, summarize, and amplify what I saw in the reviews and discussion. Miscommunication in the reviewer-author dialog is very frustrating from both the authors' and reviewer's perspectives, but I believe that the discussion was ultimately fruitful; it certainly helped me to see both the valuable core of what the authors are trying to accomplish with this paper and the constructive critique that could help the paper be more effective. I hope that the authors can find value in the feedback from this review process as they presumably look toward revising this manuscript.

**Audience:**

Yes

**Audience Explanation:**

Especially since it came up in some of the reviews, discussions, and final recommendations, I emphasize here that "novelty" and "predicted significance" are _not_ primary acceptance criteria at TMLR and, as the action editor, I am careful to discount or even disregard critiques on these dimensions, generally from reviewers used to criteria at other publication venues.

The bar here is simply that some members of the TMLR readership would be interested in the paper. Often the fact that some or all of the reviewers are interested is sufficient evidence that the answer is yes. As all three of the reviewers felt that the answer was yes for this paper, I could hardly argue otherwise. Certainly I agree that the overall topic of leveraging LLMs or other large foundation models in more autonomous, dynamic, adaptable agents would be of interest to the TMLR readership.

That said, from my own reading, I do have concerns about audience and, had I been a reviewer myself, I may very well have selected "no." Again, this comes back to a lack of clarity about what the paper is proposing. The final section says that "We provide a more
pragmatic plan as a complement [to Silver and Sutton, 2025]." But upon reaching this section I was still unsure about what that plan actually was, let alone whether it was pragmatic. It says that "We highlight the importance of both iterative improvements and ground truth,
propose to explore and exploit domain knowledge w.r.t. data collection, architecture design and algorithm
design, with decision time planning and meta RL at both pre- and post-training stages." Those are all sensible things to want in an AI agent (though perhaps not universal requirements for all problem settings or applications), but I did not feel that I had a clearer vision about how they would all come together in an agent, or how they connect to LLMs, than when I started. If I were someone with perspectives that this paper aims to counter (e.g. believing in the power of enormous, general purpose, pre-trained models), I'm not at all sure that I would have found compelling arguments to cause me to question my perspective.

So my advice to the authors for this criterion is to think carefully about what effect the perspective paper should have on the reader. Should they see a research path that they haven't seen before? Should they change their mind about something? Should they be inspired to work on a problem they might not have otherwise? Whatever it is, focus the energy of the manuscript on that purpose so the reader comes away changed somehow and glad that they have spent the time to engage with the paper. For a perspective papers, that might take the form of "You might think this or that, but that's probably not right because..." or maybe "Not many people are studying this or that problem, but it's really important because..." As written, the paper seems diffuse in its purpose. It presents an abstract proposal, but to what end? It argues for the importance of some ideas, but important for what? As such, though in my role as AE I am selecting "yes" for this question, as a reader it was not clear to me what the intended audience was for the paper or what that audience would take away from it.

**Claims And Evidence:**

No

**Claims Explanation:**

This is a challenging review criterion to apply to this paper because it is presented as a perspective paper and thus makes very few concrete or technical claims. Some of the claims it does make are speculative in nature, as is appropriate in a position/perspective paper.

That said, the paper's discussion centers on an "approach"/"proposal"/"plan"; many of the claims are about these central ideas. In their reviews and in the discussion, the reviewers grappled with the clarity of the proposal and, relatedly, the quality of the arguments for claims/predictions about the proposal. Since these matters concern the coherence and soundness of the primary contribution of the paper, I think they are appropriately considered under this category.

I have carefully considered the extensive discussion between the reviewers (reviewer Jppn in particular) and the authors. Considering the contentious nature of that exchange, I have also read the revised manuscript to ground that discussion in the text itself. Ultimately I conclude that many of the critiques raised flow from a central structural problem, namely that the actual proposal itself is not sufficiently clearly stated. The three LLM agent frameworks are lightly sketched. It is clear that the aim here is to bridge LLM training and the reinforcement learning problem, but the connection is tenuous. Often the described framework seems to be a standard RL methodology and, though LLMs are mentioned, it's not clear how the LLM itself is utilized or leveraged.

More specifically:
- AgentZero: "Stage 1" mentions self-supervised learning for representation learning. Learning the representation of what? Self-supervised using what signal? This is described as being "as in normal LLMs." Does this mean Stage 1 is about training an LLM? What would the textual corpus be? In this case the agent is assumed to have a perfect predictive model, so the LLM does not play the role of the model. What role does it play? For "Stage 2," the paper describes a typical RL agent with states, actions, and rewards learning over time to make decisions that lead to high rewards. Perhaps the LLM is meant to take the place of a value function or a policy? But text prediction and value prediction are fundamentally different learning problems. So for AgentZero it is not clear whether an LLM is involved and, if so, how it relates to the reinforcement learning agent being described.
- AgentMu: The description seems to describe a standard model-based reinforcement learning agent that learns a model of its environment and uses planning of some form to make decisions. Once again, it is not clear how an LLM is involved. The text says "The significant difference between Agentµ and most current LLM agents hinges on iterative improvements based on ground truth of the base LLM and the world model." This seems to suggest that the LLM and the world model are distinct. Presumably the "iterative improvements" are occurring within the RL component of the agent, and possibly the world model as well. From this text I can't tell what role the LLM has.
- AgentInfinity: As above, this seems to describe the standard model-free RL problem. It is not at all clear what role the LLM plays in solving the RL problem.

After discussion, all three reviewers agreed that there are areas of the paper that seem imprecise, poorly defined, or poorly supported. From my own read of the paper, I agree, and I think this is often caused by lack of a clear referent. When the rest of the paper speculates about pros and cons of the proposed approach, or posits potential challenges and possible solutions, that discussion is not grounded in a clear vision of what the proposed approach actually is.

My suggestion to the authors, in continuing to develop these ideas, is to focus in on a clear description of the framework being proposed. What are its components? How do they interact? What are the processes that operate on data? When do various processes occur (e.g. offline/pre-training vs. at decision time)? It is understood that the authors intend to describe a general framework and not a specific implementation; surely many details will be abstract. Nevertheless, in order for the reader to follow and evaluate the reasoning about the implications of the idea, the idea itself must be clear. During the discussion the authors raised concerns about space; to my eyes, the paper could afford to lose or reduce some of the long citation lists (often presented with little to no context/discussion) in exchange for a clear articulation of the central contribution.

**Resubmission Of Major Revision:**

The authors may consider submitting a major revision at a later time.

---

> ### Author Response · Authors · 2025-08-14
> **Sadly,  both Action Editor 43NQ and Reviewer Jppn do not have a good understanding of the submission.**
>
> However, Reviewer Jppn provided confusing and destructive review and discussions, while both Action Editor 43NQ and Reviewer Jppn have the power to reject the submission.
>
> Action Editor 43NQ mentioned that: "it's not clear how the LLM itself is utilized or leveraged", and asked several questions about how the LLM is involved.
>
> We made it very clear that ideally we do pre-training, so that no existing LLM is involved.
> Even if we may start with a pre-trained LLM and we do post-training, we do not rely on the LLM.
>
> All current LLMs are not perfect, so that they won't provide ground truth, wrt to a world model, a policy, a value function, data generation, evaluation/judgement, ..., everything, unless with external assistance, e.g., a maths verifier.
>
> We propose to make iterative improvements based on ground truth.
> It is in the title.

---

> > ### Comment · Action_Editor_43NQ · 2025-08-14
> >
> > Hello authors,
> >
> > This will be my last comment on this submission.
> >
> > I can't speak for Reviewer Jppn, but for myself I can confirm that I do not have a good understanding of the submission despite considerable good faith effort and extended engagement with the authors' attempts to clarify. This is precisely the central concern raised by the reviewers (not just Reviewer Jppn) that I tried to capture in my meta-review. Understanding is a two way process and it can falter at both the reading and the writing ends. The reviewing team on this paper has background, expertise, and active interest in your topic; we are all your audience and we have read your paper carefully and offered detailed critique, which is more than most of your audience would be expected to do. So there are two possibilities here: either the paper is clear, precise, and easily understood except to this particular reviewing team, or the paper does not fully support an expert, motivated reader in understanding its core message or contributions.
> >
> > For example, in the authors' comment here they emphasize that their proposed framework would involve training novel LLMs and that they "propose to make iterative improvements based on ground truth," with the snarky addition that "it is in the title." I have indeed read the title. The clarity on where the LLM will come from does address a minor question that I had, but this comment, seemingly intended to demonstrate that all of my questions are straightforwardly answered by the paper, does not seem to engage with my central question of what the proposed agent frameworks actually are and how the LLM (however it is trained) interacts with the other components, including how the "iterative improvement" is incorporated. It is possible that my main point was not understood and, of course, that is partly on me. I tried to be clear, targeted, and concrete, but perhaps I didn't entirely succeed.
> >
> > All of which is to say that all authors inevitably find the meaning of their own writing to be clear and obvious. One of the purposes of the peer review process is to push the work outside of that bubble so that it can be refined. The decision on this paper is already made so I write this only in in the name of mentorship from scientist to scientist: you can learn something from everyone who is kind enough to read and comment on your paper, but it requires the humility and the growth mindset to imagine that something about the paper could be improved, even if it is just to better emphasize and clarify points that you believe you have already made (I'm not convinced that this is the case here, but we can agree to disagree). We've all felt righteous indignation about a frustrating review process at times, but in the long run you'll gain much more value from the peer review process by focusing less on perceived flaws in the feedback you are given and more on how the feedback can help you be a better scientist and a better communicator. I still sincerely hope that you will be able to find some such value from this process.
> >
> > I hope that this is received as it is intended, some good faith feedback on the authors' role in the process and some sincere advice from an increasingly old timer who has learned these lessons the hard way. In any case, I have no intention of engaging in a debate about the decision (which is finalized) or the legitimacy of the process (which I believe to be sufficiently intact), so I do not plan to respond any further. My best wishes to the authors and their future endeavors.

---

> > > ### Author Response · Authors · 2025-08-14
> > >
> > > If the reviewers and editor reject a submission based on misunderstandings of the submission, what the authors should learn?

---

> > > > ### Comment · Reviewer_Jppn · 2025-08-14
> > > >
> > > > One thing to learn is to revise the paper in a way that prevents misunderstandings of the submission in the next version. Or, if the authors feel they are mistreated, please feel free to report to editors-in-chief as suggested on the TMLR website.

---

> > > > > ### Author Response · Authors · 2025-08-14
> > > > >
> > > > > Jppn: You do not have enough background to review this submission as complained.

---

> > > > > > ### Author Response · Authors · 2025-08-14
> > > > > >
> > > > > > The first author suffered a lot from Reviewer Jppn, who lacks expertise and professionalism.
> > > > > >
> > > > > > We thought the Action Editor would make a fair decision.
> > > > > > BUT, the Action Editor misunderstood the submission, and blamed the authors' writing.
> > > > > >
> > > > > > Will Editor-in-Chief help? haha
> > > > > >
> > > > > > The lesson learned: give up, no more waste of time, no more suffering.